# Efficient Projection-Free Algorithms for Saddle Point Problems

**Cheng Chen**[1] **Luo Luo**[2*] **Weinan Zhang**[1] **Yong Yu**[1]
[1]Shanghai Jiao Tong University
[2]The Hong Kong University of Science and Technology
jack_chen1990@sjtu.edu.cn    luoluo@ust.hk    {wnzhang,yyu}@apex.sjtu.edu.cn

## Abstract

The Frank-Wolfe algorithm is a classic method for constrained optimization problems. It has recently been popular in many machine learning applications because its projection-free property leads to more efficient iterations. In this paper, we study projection-free algorithms for convex-strongly-concave saddle point problems with complicated constraints. Our method combines Conditional Gradient Sliding with Mirror-Prox and shows that it only requires $\tilde{\mathcal{O}}(1/\sqrt{\epsilon})$ gradient evaluations and $\tilde{\mathcal{O}}(1/\epsilon^2)$ linear optimizations in the batch setting. We also extend our method to the stochastic setting and propose first stochastic projection-free algorithms for saddle point problems. Experimental results demonstrate the effectiveness of our algorithms and verify our theoretical guarantees.

## 1 Introduction

In this paper, we study the following saddle point problems:

$$\min_{\mathbf{x} \in \mathcal{X}} \max_{\mathbf{y} \in \mathcal{Y}} f(\mathbf{x}, \mathbf{y})$$

where the objective function $f(\mathbf{x}, \mathbf{y})$ is convex-concave and $L$-smooth; $\mathcal{X}$ and $\mathcal{Y}$ are convex and compact sets. Besides this general form, we also consider the stochastic minimax problem:

$$\min_{\mathbf{x} \in \mathcal{X}} \max_{\mathbf{y} \in \mathcal{Y}} f(\mathbf{x}, \mathbf{y}) \triangleq \mathbb{E}_\xi [F(\mathbf{x}, \mathbf{y}; \xi)], \tag{1}$$

where $\xi \in \Xi$ is a random variable. One popular specific setting of (1) is the finite-sum case where $\xi$ is sampled from a finite set $\Xi = \{\xi_i\}_{i=1}^n$. Denoting $F_i(\mathbf{x}, \mathbf{y}) \triangleq F(\mathbf{x}, \mathbf{y}; \xi_i)$, we can write the objective function as

$$f(\mathbf{x}, \mathbf{y}) \triangleq \frac{1}{n} \sum_{i=1}^n F_i(\mathbf{x}, \mathbf{y}). \tag{2}$$

We are interested in the cases where the feasible set is complicated such that projecting onto $\mathcal{X} \times \mathcal{Y}$ is rather expensive or even intractable. One example of such case is the nuclear norm ball constraint, which is widely used in machine learning applications such as multiclass classification [5], matrix completion [2, 14, 17], factorization machine [20], polynomial neural nets [21] and two-player games whose strategy space contains a large number of constraints [1].

The Frank-Wolfe (FW) algorithm [6] (a.k.a. conditional gradient method) is initially proposed for constrained convex optimization. It has recently become popular in the machine learning community

---

because of its projection-free property [13]. The Frank-Wolfe algorithm calls a linear optimization (LO) oracle at each iteration, which is usually much faster than projection for complicated feasible sets. Recently, FW-style algorithms for convex and nonconvex minimization problems has been widely studied [9, 10, 11, 17, 19, 27, 28, 30, 32, 33, 34]. However, the only known projection-free algorithms for minimax optimization are for very special cases (e.g. the saddle point belongs to the interior of the feasible set [7]).

In this paper, we propose a projection-free algorithm, which we refer to as Mirror-Prox Conditional Gradient Sliding (MPCGS), for convex-strongly-concave saddle point problems. Our method leverages the idea from some projection-type methods [23, 31], which is based on proximal point iterations. By combining the idea of Mirror-Prox [31] with the conditional gradient sliding (CGS) [19], MPCGS only requires at most $\tilde{\mathcal{O}}(1/\sqrt{\epsilon})$ exact gradient evaluations and $\tilde{\mathcal{O}}(1/\epsilon^2)$ linear optimizations to guarantee $\mathcal{O}(\epsilon)$ suboptimality error in expectation. We also extend our framework to the stochastic setting and propose Mirror-Prox Stochastic Conditional Gradient Sliding (MPSCGS), which requires to compute at most $\tilde{\mathcal{O}}(1/\epsilon^2)$ stochastic gradients and call the LO oracle for at most $\tilde{\mathcal{O}}(1/\epsilon^2)$ times. To the best of our knowledge, MPSCGS is the first stochastic projection-free algorithm for convex-strongly-concave saddle point problems. We also conduct experiments on several real-world data sets for robust optimization problem to validate our theoretical analysis. The empirical results show that the proposed methods outperform previous projection-free and projection-based methods when the feasible set is complicated.

**Related Works**  Most existing works on constrained minimax optimization solve the problem with projection. We only provide some representative literature. For the batch setting, the classical extragradient method [16] considered a more general variational inequality (VI). Nemirovski [23] proposed Mirror-Prox method which achieves a convergence rate of $\mathcal{O}(1/\epsilon)$ for solving VI. Recently, Thekumparampil et al. [31] improved the convergence rate to $\mathcal{O}(1/\sqrt{\epsilon})$ when the objection function is strongly-convex-concave. For the stochastic setting, Chavdarova et al. [3], Palaniappan and Bach [26] adopted the variance reduction methods to obtain linear convergence rate for strongly-convex-strongly-concave objection functions.

The projection-free methods for saddle point problems are very few. Hammond [8] found that FW algorithm with a step size $\mathcal{O}(1/k)$ converges for VI when the feasible set is strongly convex. Recently, Gidel et al. [7] proposed SP-FW algorithm for strongly-convex-strongly-concave saddle point problem, which achieves a linear convergence rate under the condition that the saddle point belongs to the interior of the feasible set and the condition number is small enough. They also provided an away-step Frank-Wolfe variant [17], called SP-AFW, to address the polytope constraints. However, SP-AFW has to store history information and perform extra operations in each iteration. Roy et al. [29] extended SP-FW to the zeroth-order setting and studied a gradient-free projection-free algorithm which has the theoretical guarantee under the same assumptions on the objective function as SP-FW. He and Harchaoui [12] proposed a projection-free algorithm for non-smooth composite saddle point problem. Their method requires to call a composite LO oracle, which is not suitable for general case.

Some recent works focus on hybrid algorithms which combine projection-based and projection-free methods. For example, Cox et al. [4], Juditsky and Nemirovski [15] transformed a VI with complicated constraints to a "dual" VI which is projection-friendly. Lan [18], Nouiehed et al. [25] solved the saddle point problem by running projection-free methods on $\mathcal{X}$ and performing projection on $\mathcal{Y}$. In contrast, our methods are purely projection-free.

**Paper Organization**  In Section 2, we provide preliminaries and relevant backgrounds. We present our results for the batch setting and stochastic setting in Section 3 and Section 4 respectively. We give empirical results for our algorithm in Section 5, followed by a conclusion in Section 6.

## 2   Preliminaries and Backgrounds

In this section, we first present some notation and assumptions used in this paper. Then we introduce oracle models which are necessary to our methods, followed by an example of application. After that we provide some properties of saddle point problems. Finally, we introduce CGS and its variants, which are used in our algorithms.

## 2.1 Notation and Assumptions

Given a differentiable function $f(\mathbf{x}, \mathbf{y})$, we use $\nabla_{\mathbf{x}} f(\mathbf{x}, \mathbf{y})$ (or $\nabla_{\mathbf{y}} f(\mathbf{x}, \mathbf{y})$) to denote the partial gradient of $f$ with respect to $\mathbf{x}$ (or $\mathbf{y}$) and define $f_{\mathcal{S}}(\mathbf{x}, \mathbf{y}) = \frac{1}{|\mathcal{S}|} \sum_{\xi \in \mathcal{S}} F(\mathbf{x}, \mathbf{y}; \xi)$. We use the notation $\tilde{\mathcal{O}}$ to hide logarithmic factors in the complexity and denote $[n] = \{1, 2, \ldots, n\}$.

We impose the following assumptions for our method.

**Assumption 1.** *We assume the saddle point problem (1) satisfies:*

- $f(\mathbf{x}, \mathbf{y})$ *is L-smooth, i.e., for every* $(\mathbf{x}_1, \mathbf{y}_1), (\mathbf{x}_2, \mathbf{y}_2) \in \mathcal{X} \times \mathcal{Y}$*, it holds that*
$$\|\nabla f(\mathbf{x}_1, \mathbf{y}_1) - \nabla f(\mathbf{x}_2, \mathbf{y}_2)\|^2 \leq L^2 \left( \|\mathbf{x}_1 - \mathbf{x}_2\|^2 + \|\mathbf{y}_1 - \mathbf{y}_2\|^2 \right).$$

- $f(\cdot, \mathbf{y})$ *is convex for every* $\mathbf{y} \in \mathcal{Y}$*, i.e., for any* $\mathbf{x}_1$*,* $\mathbf{x}_2$ *and* $\mathbf{y}$*, it holds that*
$$f(\mathbf{x}_1, \mathbf{y}) - f(\mathbf{x}_2, \mathbf{y}) \geq \nabla_{\mathbf{x}} f(\mathbf{x}_2, \mathbf{y})^{\top} (\mathbf{x}_1 - \mathbf{x}_2).$$

- $f(\mathbf{x}, \cdot)$ *is* $\mu$*-strongly concave for every* $\mathbf{x} \in \mathcal{X}$*, i.e., for any* $\mathbf{x}$*,* $\mathbf{y}_1$ *and* $\mathbf{y}_2$*, it holds that*
$$f(\mathbf{x}, \mathbf{y}_1) - f(\mathbf{x}, \mathbf{y}_2) \leq \nabla_{\mathbf{y}} f(\mathbf{x}, \mathbf{y}_2)^{\top} (\mathbf{y}_1 - \mathbf{y}_2) - \frac{\mu}{2} \|\mathbf{y}_1 - \mathbf{y}_2\|^2.$$

- $\mathcal{X}$ *and* $\mathcal{Y}$ *are convex compact sets with diameter* $D_{\mathcal{X}}$ *and* $D_{\mathcal{Y}}$ *respectively.*

We use $\kappa = L/\mu$ to denote the condition number.

**Assumption 2.** *In the stochastic setting, we make the following additional assumptions:*

- $\mathbb{E}[\nabla F(\mathbf{x}, \mathbf{y}; \xi)] = \nabla f(\mathbf{x}, \mathbf{y})$ *for every* $(\mathbf{x}, \mathbf{y}) \in \mathcal{X} \times \mathcal{Y}$ *and* $\xi \in \Xi$*.*
- $\mathbb{E}\|\nabla F(\mathbf{x}, \mathbf{y}; \xi) - \nabla f(\mathbf{x}, \mathbf{y})\|^2 \leq \sigma^2$ *for every* $(\mathbf{x}, \mathbf{y}) \in \mathcal{X} \times \mathcal{Y}$*,* $\xi \in \Xi$ *and constant* $\sigma > 0$*.*
- $f(\mathbf{x}, \mathbf{y})$ *is L-average smooth, i.e., for every* $(\mathbf{x}_1, \mathbf{y}_1), (\mathbf{x}_2, \mathbf{y}_2) \in \mathcal{X} \times \mathcal{Y}$ *and* $\xi \in \Xi$*, it holds that*
$$\mathbb{E}\|\nabla F(\mathbf{x}_1, \mathbf{y}_1, \xi) - \nabla F(\mathbf{x}_2, \mathbf{y}_2, \xi)\|^2 \leq L^2 (\|\mathbf{x}_1 - \mathbf{x}_2\|^2 + \|\mathbf{y}_1 - \mathbf{y}_2\|^2).$$

In the convex-concave setting, for any $(\hat{\mathbf{x}}, \hat{\mathbf{y}}) \in \mathcal{X} \times \mathcal{Y}$, we have the following inequality:
$$\min_{\mathbf{x} \in \mathcal{X}} f(\mathbf{x}, \hat{\mathbf{y}}) \leq f(\hat{\mathbf{x}}, \hat{\mathbf{y}}) \leq \max_{\mathbf{y} \in \mathcal{Y}} f(\hat{\mathbf{x}}, \mathbf{y}).$$

Furthermore, Problem (1) has at least one saddle point solution $(\mathbf{x}^*, \mathbf{y}^*) \in \mathcal{X} \times \mathcal{Y}$ which satisfies:
$$\min_{\mathbf{x} \in \mathcal{X}} f(\mathbf{x}, \mathbf{y}^*) = f(\mathbf{x}^*, \mathbf{y}^*) = \max_{\mathbf{y} \in \mathcal{Y}} f(\mathbf{x}^*, \mathbf{y}).$$

We measure the suboptimality error by the primal-dual gap: $\max_{\mathbf{y} \in \mathcal{Y}} f(\hat{\mathbf{x}}, \mathbf{y}) - \min_{\mathbf{x} \in \mathcal{X}} f(\mathbf{x}, \hat{\mathbf{y}})$, which is widely used in saddle point problems. We further define $\epsilon$-saddle point as follows:

**Definition 1.** *A point* $(\hat{\mathbf{x}}, \hat{\mathbf{y}}) \in \mathcal{X} \times \mathcal{Y}$ *is an* $\epsilon$*-saddle point of a convex-concave function* $f$ *if:*
$$max_{\mathbf{y} \in \mathcal{Y}} f(\hat{\mathbf{x}}, \mathbf{y}) - \min_{\mathbf{x} \in \mathcal{X}} f(\mathbf{x}, \hat{\mathbf{y}}) \leq \epsilon. \tag{3}$$

Notice that Gidel et al. [7] adopted a different criterion: $w(\hat{\mathbf{x}}, \hat{\mathbf{y}}) = f(\hat{\mathbf{x}}, \mathbf{y}^*) - f(\mathbf{x}^*, \hat{\mathbf{y}})$. It is obvious that the left-hand side of (3) is an upper bound of $w(\hat{\mathbf{x}}, \hat{\mathbf{y}})$.

## 2.2 Oracle models

In this paper, we consider the following oracles for different settings:

- First Order Oracle (FO): Given $(\mathbf{x}, \mathbf{y}) \in \mathcal{X} \times \mathcal{Y}$, the FO returns $f(\mathbf{x}, \mathbf{y})$ and $\nabla f(\mathbf{x}, \mathbf{y})$.
- Stochastic First Order Oracle (SFO): For a function $\mathbb{E}_{\xi}[F(\mathbf{x}, \mathbf{y}; \xi)]$ where $\xi \sim P$, SFO returns $F(\mathbf{x}, \mathbf{y}; \xi')$ and $\nabla F(\mathbf{x}, \mathbf{y}; \xi')$ where $\xi'$ is a sample drawn from $P$.
- Incremental First Order Oracle (IFO): In the finite-sum setting, IFO takes a sample $i \in [n]$ and returns $F_i(\mathbf{x}, \mathbf{y})$ and $\nabla F_i(\mathbf{x}, \mathbf{y})$.
- Linear Optimization Oracle (LO): Given a vector $\mathbf{g} \in \mathbb{R}^d$ and a convex and compact set $\Omega \subseteq \mathbb{R}^d$, the LO returns a solution of the problem $\min_{\mathbf{v} \in \Omega} \langle \mathbf{v}, \mathbf{g} \rangle$.

---

**Algorithm 1** CGS Method for strongly convex functions

---
**Input:** $L$-smooth and $\mu$-strongly-convex function $h$, convex and compact set $\Omega$, total iterations $N$.
**Input:** The initial point $\bar{\mathbf{x}}_0 \in \Omega$ satisfies $h(\bar{\mathbf{x}}_0) - h(\mathbf{x}^*) \leq \delta_0$.

 1: $M \leftarrow \sqrt{24L/\mu}$.
 2: **for** $t = 1, 2, \ldots, N$ **do**
 3: $\quad \mathbf{x}_0 \leftarrow \bar{\mathbf{x}}_{t-1}, \mathbf{u}_0 \leftarrow \mathbf{x}_0$
 4: $\quad$ **for** $k = 1, 2, \ldots, M$ **do**
 5: $\qquad \lambda_k \leftarrow \frac{2}{k+1}, \beta_k \leftarrow \frac{2L}{k}, \eta_{t,k} \leftarrow \frac{8L\delta_0 2^{-t}}{\mu N k}$
 6: $\qquad \mathbf{w}_k \leftarrow (1 - \lambda_k)\mathbf{x}_{k-1} + \lambda_k \mathbf{u}_{k-1}$
 7: $\qquad \mathbf{u}_k \leftarrow \text{CndG}(\nabla h(\mathbf{w}_k), \mathbf{u}_{k-1}, \beta_k, \eta_{t,k}, \Omega)$
 8: $\qquad \mathbf{x}_k \leftarrow (1 - \lambda_k)\mathbf{x}_{k-1} + \lambda_k \mathbf{u}_k$
 9: $\quad$ **end for**
 10: $\quad \bar{\mathbf{x}}_t = \mathbf{x}_M$
 11: **end for**

---

### 2.3 Example Application: Robust Optimization for Multiclass Classification

We consider the multiclass classification problem with $h$ classes. Suppose the training set is $\mathcal{D} = \{(\mathbf{a}_i, b_i)\}_{i=1}^n$, where $\mathbf{a}_i \in \mathbb{R}^d$ is the feature vector of the $i$-th sample and $b_i \in [h]$ is the corresponding label. The goal is to find an accurate linear predictor with parameter $\mathbf{X} = [\mathbf{x}_1^\top, \mathbf{x}_2^\top, \cdots, \mathbf{x}_h^\top] \in \mathbb{R}^{h \times d}$ that predicts $b = \arg\max_{j \in [h]} \mathbf{x}_j^\top \mathbf{a}$ for any input feature vector $\mathbf{a} \in \mathbb{R}^d$.

The robust optimization model [22] with multivariate logistic loss [5, 35] under nuclear norm ball constraint can be formulated as the following convex-concave minimax optimization:

$$\min_{\mathbf{X} \in \mathcal{X}} \max_{\mathbf{y} \in \mathcal{Y}} f(\mathbf{X}, \mathbf{y}) \triangleq \frac{1}{n} \sum_{i=1}^n y_i \log\left(1 + \sum_{j \neq y_i} \left(\mathbf{x}_j^\top \mathbf{a}_i - \mathbf{x}_{y_i}^\top \mathbf{a}_i\right)\right) - \frac{\lambda}{2}\|n\mathbf{y} - \mathbf{1}_n\|_2^2, \quad (4)$$

where $\mathcal{X} = \left\{\mathbf{X} \in \mathbb{R}^{h \times d} : \|\mathbf{X}\|_* \leq \tau\right\}$ and $\mathcal{Y} = \{\mathbf{y} \in \mathbb{R}^n : y_i \geq 0, \sum_{i=1}^n y_i = 1\}$. It is obvious that the objective function (4) is convex-strongly-concave, which satisfies our assumptions. In this case, projecting onto $\mathcal{X}$ requires to perform full SVD, which takes $\mathcal{O}(hd\min\{h, d\})$ time. On the other hand, the linear optimization on $\mathcal{X}$ only needs to find the top singular vector, whose cost is linear to the number of non-zero entries in the gradient matrix.

### 2.4 Conditional Gradient Sliding

Conditional Gradient Sliding (CGS) [19] is a projection-free algorithm for convex minimization. It leverages Nesterov's accelerate gradient descent [24] to speed-up Frank-Wolfe algorithms. For strongly-convex objective function, CGS only requires $\mathcal{O}(\sqrt{\kappa}\log(1/\epsilon))$ FO calls and $\mathcal{O}(1/\epsilon)$ LO calls to find an $\epsilon$-suboptimal solution. We present the details of CGS in Algorithm 1. Notice that the $k$-th iteration of CGS considers the following sub-problem

$$\min_{\mathbf{u} \in \Omega} \langle \nabla h(\mathbf{w}_k), \mathbf{u}\rangle + \frac{\beta_k}{2}\|\mathbf{u} - \mathbf{u}_{k-1}\|^2,$$

which can be efficiently solved by the conditional gradient method in Algorithm 2. Lan and Zhou [19] also extended CGS to stochastic setting and proposed stochastic conditional gradient sliding (SCGS). Later, Hazan and Luo [11] proposed STOchastic variance-Reduced Conditional gradient sliding (STORC) for finite-sum setting whose complexities of IFO and LO are $\mathcal{O}((n + \kappa^2)\log(1/\epsilon))$ and $\mathcal{O}(1/\epsilon)$ respectively.

## 3 Mirror-Prox Conditional Gradient Sliding

For the batch setting of (1), we propose Mirror-Prox Conditional Gradient Sliding (MPCGS), which is presented in Algorithm 3. Our MPCGS method combines ideas of Mirror-Prox algorithm [31] and CGS method [19]. The key idea of MPCGS is to solve a proximal problem in each iteration, which makes $\mathbf{x}_k$ and $\mathbf{y}_k$ satisfy following conditions:

---

**Algorithm 2** Procedure $\mathbf{q}^+ = \text{CndG}(\mathbf{r}, \mathbf{q}, \beta, \eta, \Omega)$

---

1: $\mathbf{q}_1 \leftarrow \mathbf{q}$.
2: **for** $t = 1, 2, \ldots$ **do**
3: $\quad \mathbf{p}_t \leftarrow \arg\max_{\mathbf{x} \in \Omega} \langle \mathbf{r} + \beta(\mathbf{q}_t - \mathbf{q}), \mathbf{q}_t - \mathbf{x} \rangle$, $\tau_t \leftarrow \langle \mathbf{r} + \beta(\mathbf{q}_t - \mathbf{q}), \mathbf{q}_t - \mathbf{p}_t \rangle$
4: $\quad$ If $\tau_t \leq \eta$, set $\mathbf{q}^+ = \mathbf{q}_t$ and terminate the procedure.
5: $\quad \theta_t \leftarrow \min\left\{1, \frac{\tau_t}{\beta\|\mathbf{q}_t - \mathbf{p}_t\|^2}\right\}$, $\mathbf{q}_{t+1} \leftarrow (1 - \theta_t)\mathbf{q}_t + \theta_t \mathbf{p}_t$
6: **end for**

---

- $\mathbf{y}_k$ is an $\epsilon_k$-approximate maximizer of $f(\mathbf{x}_k, \cdot)$, i.e., $f(\mathbf{x}_k, \mathbf{y}_k) \geq \max_{\mathbf{y}} f(\mathbf{x}_k, \mathbf{y}) - \epsilon_k$;
- The update of $\mathbf{x}_k$ corresponds to an CGS updating step (Algorithm 1) for $-f(\cdot, \mathbf{y}_k)$, i.e.,

$$\mathbf{v}_k = \text{CndG}(\nabla_{\mathbf{x}} f(\mathbf{z}_k, \mathbf{y}_k), \mathbf{v}_{k-1}, \alpha_k, \zeta_k, \mathcal{X}), \quad \mathbf{x}_k = (1 - \gamma_k)\mathbf{x}_{k-1} + \gamma_k \mathbf{v}_k.$$

The procedure of solving the proximal problem is presented in Algorithm 4. In the Prox-step procedure, we iteratively compute an $\epsilon_{cgs}$-approximate maximizer of $f(\mathbf{x}_{r-1}, \cdot)$ and then update $\mathbf{v}_r$ and $\mathbf{x}_r$ according to $\mathbf{y}_r$. Since $f(\mathbf{x}, \cdot)$ is smooth and strongly concave for all $\mathbf{x} \in \mathcal{X}$, the number of calls to the FO and LO oracles performed by CGS method for finding an $\epsilon_k$-approximate maximizer can be bounded by $\mathcal{O}(\sqrt{\kappa} \log(1/\epsilon_{cgs}))$ and $\mathcal{O}(1/\epsilon_{cgs})$ respectively.

On the other hand, in Algorithm 4 the CndG procedure computes $\mathbf{v}_r$ as an $\zeta$-approximate solution of the following problem:

$$\min_{\mathbf{u} \in \mathcal{X}} \left\{ \nabla_{\mathbf{x}} f(\mathbf{z}, \mathbf{y}^*(\mathbf{x}_{r-1}))^\top \mathbf{u} + \frac{\alpha}{2}\|\mathbf{u} - \mathbf{v}\|^2 \right\}.$$

Thus, the idealized updating of $\mathbf{x}_r$ in Algorithm 4 is

$$\mathbf{x}_r = (1 - \gamma)\mathbf{x} + \gamma \cdot \arg\min_{\mathbf{u} \in \mathcal{X}} \left\{ \nabla_{\mathbf{x}} f(\mathbf{z}, \mathbf{y}^*(\mathbf{x}_{r-1}))^\top \mathbf{u} + \frac{\alpha}{2}\|\mathbf{u} - \mathbf{v}\|^2 \right\}.$$

Since $\psi(\mathbf{x}) \triangleq (1-\gamma)\mathbf{x} + \gamma \cdot \arg\min_{\mathbf{u} \in \mathcal{X}}\{\nabla_{\mathbf{x}} f(\mathbf{z}, \mathbf{y}^*(\mathbf{x}_{r-1}))^\top \mathbf{u} + \frac{\alpha}{2}\|\mathbf{u} - \mathbf{v}\|^2\}$ is a $(1/2)$-contraction mapping with a unique fixed point (see the proof of Lemma 2 in the Appendix), the Prox-step procedure only requires $\mathcal{O}(\log(1/\epsilon))$ iterations if $\epsilon_{cgs}$ and $\zeta$ are small enough.

The following theorem shows the convergence rate of solving problem (1) by Algorithm 3.

**Theorem 1.** *Suppose the objective function $f(\mathbf{x}, \mathbf{y})$ satisfies Assumption 1. By setting*

$$\gamma_k = \frac{3}{k+2}, \quad \alpha_k = \frac{6\kappa L}{k+1}, \quad \zeta_k = \frac{LD_{\mathcal{X}}^2}{384k(k+1)}, \quad \epsilon_k = \frac{\kappa L D_{\mathcal{X}}^2}{k(k+1)(k+2)}$$

*for Algorithm 3, then we have*

$$\max_{\mathbf{y} \in \mathcal{Y}} f(\mathbf{x}_k, \mathbf{y}) - \min_{\mathbf{x} \in \mathcal{X}} f(\mathbf{x}, \bar{\mathbf{y}}_k) \leq \frac{11\kappa L D_{\mathcal{X}}^2}{(k+1)(k+2)}.$$

Theorem 1 implies the upper bound complexities of the algorithm as follows.

**Corollary 1.** *Under the same assumption of Theorem 1, Algorithm 3 requires $\tilde{\mathcal{O}}\left(\kappa\sqrt{LD_{\mathcal{X}}^2/\epsilon}\right)$ FO complexity and $\tilde{\mathcal{O}}\left(\kappa^2 L^2 D_{\mathcal{X}}^2 D_{\mathcal{Y}}^2/\epsilon^2\right)$ LO complexity to achieve an $\epsilon$-saddle point.*

## 4 Mirror-Prox Stochastic Conditional Gradient Sliding

In this section, we extend MPCGS to the stochastic setting (1). Recall that we adopt CGS to find $\epsilon$-approximate maximizer of problem $\max_{\mathbf{y} \in \mathcal{Y}} f(\mathbf{x}_k, \mathbf{y})$ in the batch setting, which only require logarithmic iterations. In the stochastic case, we would like to use the STORC [11] algorithm instead. Since the original STORC can only be applied to the finite-sum situation, we have to first study an inexact variant of STORC which does not depend on the exact gradient. Then we leverage the inexact STORC algorithm to establish our projection-free algorithm for stochastic saddle point problems.

---

**Algorithm 3** Mirror-Prox Conditional Gradient Sliding

---

**Input:** Objective function $f(\mathbf{x}, \mathbf{y})$, parameters $\gamma_k, \alpha_k, N, \mathbf{x}_0, \mathbf{y}_0, \epsilon_k$ and $\zeta_k$.
**Output:** $\mathbf{x}_N, \bar{\mathbf{y}}_N$.

1: $\mathbf{v}_0 \leftarrow \mathbf{x}_0$
2: **for** $k = 1, 2, \ldots, N$ **do**
3:      $\mathbf{z}_k \leftarrow (1 - \gamma_k)\mathbf{x}_{k-1} + \gamma_k \mathbf{v}_{k-1}$.
4:      $(\mathbf{x}_k, \mathbf{y}_k, \mathbf{v}_k) \leftarrow$ Prox-step$(f, \mathbf{x}_{k-1}, \mathbf{y}_{k-1}, \mathbf{z}_k, \mathbf{v}_{k-1}, \gamma_k, \alpha_k, \zeta_k, \epsilon_k)$.
5:      $\bar{\mathbf{y}}_k \leftarrow \frac{3}{k(k+1)(k+2)} \sum_{s=1}^{k} s(s+1)\mathbf{y}_s$.
6: **end for**

---

---

**Algorithm 4** Procedure $(\mathbf{x}_R, \mathbf{y}_R, \mathbf{v}_R)$=Prox-step$(f, \mathbf{x}_0, \mathbf{y}_0, \mathbf{z}, \mathbf{v}, \gamma, \alpha, \zeta, \epsilon)$

---

1: $\epsilon_{cgs} \leftarrow \epsilon/(64\kappa)$, $\epsilon_{mp} \leftarrow 4\gamma\sqrt{2\kappa L\epsilon_{cgs}/\alpha^2 + 2\zeta/\alpha}$, $R \leftarrow \lceil \log_2(4D_{\mathcal{X}}/\epsilon_{mp}) \rceil$.
2: **for** $r = 1, \ldots, R$ **do**
3:      Use CGS method (Algorithm 1) with objective function $-f(\mathbf{x}_{r-1}, \cdot)$ and start point $\mathbf{y}_0$ to compute $\mathbf{y}_r$ such that: $f(\mathbf{x}_{r-1}, \mathbf{y}_r) \geq \max_{\mathbf{y} \in \mathcal{Y}} f(\mathbf{x}_{r-1}, \mathbf{y}) - \epsilon_{cgs}$
4:      $\mathbf{v}_r \leftarrow$ CndG$(\nabla_{\mathbf{x}} f(\mathbf{z}, \mathbf{y}_r), \mathbf{v}, \alpha, \zeta, \mathcal{X})$,    $\mathbf{x}_r \leftarrow (1 - \gamma)\mathbf{x}_0 + \gamma\mathbf{v}_r$
5: **end for**

---

## 4.1 Inexact Stochastic Variance Reduced Conditional Gradient Sliding

We propose Inexact STORC (iSTORC) algorithm to solve the following stochastic convex optimization problem:

$$\min_{\mathbf{x} \in \Omega} h(\mathbf{x}) = \mathbb{E}_\xi [H(\mathbf{x}; \xi)], \tag{5}$$

where $\xi \in \Xi$ is a random variable; the feasible set $\Omega$ is convex, compact and has diameter $D$. We assume that $h(\mathbf{x})$ is $L$-smooth and $\mu$-strongly convex. We also suppose that the algorithm can access the stochastic gradient $H(\mathbf{x}; \xi)$ which satisfies:

- $\mathbb{E}[\nabla H(\mathbf{x}; \xi)] = \nabla h(\mathbf{x}), \quad \forall \mathbf{x} \in \Omega, \forall \xi \in \Xi$.

- $\mathbb{E}[\|\nabla H(\mathbf{x}; \xi) - \nabla h(\mathbf{x})\|^2] \leq \sigma^2, \quad \forall \mathbf{x} \in \Omega, \forall \xi \in \Xi$.

- $\mathbb{E}[\|\nabla H(\mathbf{x}_1; \xi) - \nabla H(\mathbf{x}_2; \xi)\|^2] \leq L^2 \|\mathbf{x}_1 - \mathbf{x}_2\|^2. \quad \forall (\mathbf{x}_1, \mathbf{x}_2) \in \Omega^2, \forall \xi \in \Xi$

The idea of iSTORC is to approximate the exact gradient in STORC by appropriate number of stochastic gradient samples. The following theorem shows the convergence rate of iSTORC.

**Theorem 2.** *Running Inexact STORC (Algorithm 5) with the following parameters:*

$$\lambda_k = \frac{2}{k+1}, \beta_k = \frac{3L}{k}, M = \left\lceil 4\sqrt{2\kappa} \right\rceil, \eta_{t,k} = \frac{\kappa L D^2}{2^{t-2}Mk}, S = 4800M\kappa, Q_t = \left\lceil \frac{1200 \cdot 2^{t-1}\sigma^2\sqrt{\kappa}}{L^2 D^2} \right\rceil,$$

*we have*

$$\mathbb{E}[h(\bar{\mathbf{x}}_t) - h(\mathbf{x}^*)] \leq \frac{LD^2}{2^{t+1}},$$

*where* $\mathbf{x}^* \in \arg\min_{\mathbf{x} \in \Omega} h(\mathbf{x})$.

Theorem 2 implies the following upper bound complexities of iSTORC.

**Corollary 2.** *To achieve* $\bar{\mathbf{x}}_t$ *such that* $\mathbb{E}[h(\bar{\mathbf{x}}_t) - h(\mathbf{x}^*)] \leq \epsilon$, *iSTORC (Algorithm 5) requires* $\mathcal{O}\left((\sqrt{\kappa}/(L\epsilon) + \kappa^2)\log(LD^2/\epsilon)\right)$ *SFO complexity and* $\mathcal{O}\left(LD^2/\epsilon\right)$ *LO complexity.*

**Remark 1.** *If the objective function has the finite-sum form, we can choose* $\mathcal{Q}_t = \{\xi_1, \ldots, \xi_n\}$ *and obtain the same upper complexities bound as STORC.*

**Remark 2.** *Notice that the SFO complexity of SCGS is* $\mathcal{O}(\kappa/(L\epsilon))$. *When* $2^{-\sqrt{\kappa}} < \epsilon < \kappa^{-1.5}$, *iSTORC has better SFO complexity than SCGS.*

**Algorithm 5** Inexact STORC (iSTORC)

---

**Input:** $L$-smooth and $\mu$-strongly convex function $h(\mathbf{x})$, initial point $\bar{\mathbf{x}}_0 \in \Omega$ and total iteration $N$.
**Input:** Parameters $\gamma_k$, $\beta_k$, $Q_t$, $S$, $M$ and $\eta_{t,k}$.

1: **for** $t = 1, 2, \ldots, N$ **do**
2:     $\mathbf{x}_0 \leftarrow \bar{\mathbf{x}}_{t-1}$, $\mathbf{u}_0 \leftarrow \mathbf{x}_0$.
3:     Draw $Q_t$ samples $\mathcal{Q}_t = \{\xi_j\}_{j=1}^{Q_t}$, and compute $\boldsymbol{\nu} \leftarrow \nabla h_{\mathcal{Q}_t}(\mathbf{x}_0)$.
4:     **for** $k = 1, 2, \ldots, M$ **do**
5:         $\mathbf{w}_k \leftarrow (1 - \lambda_k)\mathbf{x}_{k-1} + \lambda_k \mathbf{u}_{k-1}$.
6:         Draw $S$ samples $\mathcal{S}_{t,k} = \{\xi_{t,k,j}\}_{j=1}^{S}$ and compute $\mathbf{r}_k \leftarrow \nabla h_{\mathcal{S}_{t,k}}(\mathbf{w}_k) - \nabla h_{\mathcal{S}_{t,k}}(\mathbf{x}_0) + \boldsymbol{\nu}$.
7:         $\mathbf{u}_k \leftarrow$ CndG$(\mathbf{r}_k, \mathbf{u}_{k-1}, \beta_k, \eta_{t,k}, \Omega)$, $\mathbf{x}_k \leftarrow (1 - \lambda_k)\mathbf{x}_{k-1} + \lambda_k \mathbf{u}_k$.
8:     **end for**
9:     $\bar{\mathbf{x}}_t \leftarrow \mathbf{x}_M$.
10: **end for**

---

### 4.2 Mirror-Prox Stochastic Conditional Gradient Sliding

We present our Mirror-Prox Stochastic Conditional Gradient Sliding (MPSCGS) in Algorithm 6. The idea of MPSCGS is similar to that of MPCGS. The main difference is that we solve the proximal problem in MPSCGS by a stochastic proximal step, where we adopt the proposed iSTORC algorithm. Specifically, in each iteration we ensures that $\mathbf{x}_k$ and $\mathbf{y}_k$ satisfy following conditions:

- $\mathbf{y}_k$ is an $\epsilon_k$-approximate maximizer of $f(\mathbf{x}_k, \cdot)$ in expectation, i.e.,

$$\mathbb{E}[f(\mathbf{x}_k, \mathbf{y}_k)] \geq \mathbb{E}[\max_{\mathbf{y} \in \mathcal{Y}} f(\mathbf{x}_k, \mathbf{y})] - \epsilon_k;$$

- The update of $\mathbf{x}_k$ and $\mathbf{v}_k$ ensures that

$$\mathbf{v}_k = \text{CndG}(\nabla_{\mathbf{x}} f_{\mathcal{P}_k}(\mathbf{z}_k, \mathbf{y}_k), \mathbf{v}_{k-1}, \alpha_k, \zeta_k, \mathcal{X}), \quad \mathbf{x}_k = (1 - \gamma_k)\mathbf{x}_{k-1} + \gamma_k \mathbf{v}_k.$$

The following theorem shows the convergence rate of solving problem (1).

**Theorem 3.** *Suppose the objective function $f(\mathbf{x}, \mathbf{y})$ satisfies Assumption 1 and 2. If we set*

$$\gamma_k = \frac{3}{k+2}, \quad \alpha_k = \frac{6\kappa L}{k+1}, \quad \zeta_k = \frac{LD_{\mathcal{X}}^2}{576k(k+1)}, \quad \epsilon_k = \frac{\kappa L D_{\mathcal{X}}^2}{k(k+1)(k+2)}, \quad P_k = \left\lceil \frac{96\sigma^2(k+1)^3}{\kappa L^2 D_{\mathcal{X}}^2} \right\rceil$$

*for Algorithm 6, then we have*

$$\mathbb{E}\left[\max_{\mathbf{y} \in \mathcal{Y}} f(\mathbf{x}_k, \mathbf{y}) - \min_{\mathbf{x} \in \mathcal{X}} f(\mathbf{x}, \bar{\mathbf{y}}_k)\right] \leq \frac{12\kappa L D_{\mathcal{X}}^2}{(k+1)(k+2)}.$$

Theorem 3 implies the following corollary of oracle complexity.

**Corollary 3.** *Under the assumption in Theorem 3 and the assumption that objective function has finite-sum form of (2), Algorithm 6 needs $\tilde{\mathcal{O}}\big((n + \kappa^2)\sqrt{\kappa L D_{\mathcal{X}}^2/\epsilon} + \kappa \sigma^2 D_{\mathcal{X}}^2/\epsilon^2\big)$ IFO complexity and $\tilde{\mathcal{O}}\left(\kappa^2 L^2 D_{\mathcal{X}}^2 D_{\mathcal{Y}}^2/\epsilon^2\right)$ LO complexity to achieve an $\epsilon$-saddle point.*

**Corollary 4.** *Under the assumption in Theorem 3 and the assumption that objective function has the expectation form of (1), Algorithm 6 needs $\tilde{\mathcal{O}}\left(\kappa^{2.5}\sqrt{L D_{\mathcal{X}}^2/\epsilon} + (\kappa^{1.5} + \kappa\sigma^2)D_{\mathcal{X}}^2/\epsilon^2\right)$ SFO complexity and $\tilde{\mathcal{O}}\left(\kappa^2 L^2 D_{\mathcal{X}}^2 D_{\mathcal{Y}}^2/\epsilon^2\right)$ LO complexity to achieve an $\epsilon$-saddle point.*

## 5 Experiments

In this section, we empirically evaluate the performance of our methods on the robust multiclass classification problem introduced in Section 2.3. Specifically, we choose the nuclear norm ball with radius $\tau = 100$ and the regularization parameter $\lambda = 1/n$. We compare our methods with saddle point Frank-Wolfe (SPFW) [7] and stochastic variance reduce extragradient (SVRE) [3]. SPFW is a projection-free algorithm as discussed before, while SVRE is the state-of-the-art projection-based stochastic methods for saddle point problems. We conduct experiments on three real-world data

---

**Algorithm 6** Mirror-Prox Stochastic Conditional Gradient Sliding

---

**Input:** Objective function $f(\mathbf{x}, \mathbf{y})$, parameters $\gamma_k, \alpha_k, N, \mathbf{x}_0, \mathbf{y}_0, \epsilon_k, P_k$ and $\zeta_k$.
**Output:** $\mathbf{x}_N, \bar{\mathbf{y}}_N$.

1: $\mathbf{v}_0 \leftarrow \mathbf{x}_0$.
2: **for** $k = 1, 2, \ldots, N$ **do**
3:      $\mathbf{z}_k \leftarrow (1 - \gamma_k)\mathbf{x}_{k-1} + \gamma_t \mathbf{v}_{k-1}$.
4:      Draw $P_k$ samples $\mathcal{P}_k = \{\xi_j\}_{j=1}^{P_k}$.
5:      $(\mathbf{x}_k, \mathbf{y}_k, \mathbf{v}_k) \leftarrow$ Stochastic-Prox-step$(f, \mathbf{x}_{k-1}, \mathbf{y}_{k-1}, \mathbf{z}_k, \mathbf{v}_{k-1}, \gamma_k, \alpha_k, \zeta_k, \mathcal{P}_k, \epsilon_k)$.
6:      $\bar{\mathbf{y}}_k \leftarrow \frac{3}{k(k+1)(k+2)} \sum_{s=1}^{k} s(s+1)\mathbf{y}_s$.
7: **end for**

---

**Algorithm 7** Procedure $(\mathbf{x}_R, \mathbf{y}_R, \mathbf{v}_R) =$ Stochastic-Prox-step$(f, \mathbf{x}_0, \mathbf{y}_0, \mathbf{z}, \mathbf{v}, \gamma, \alpha, \zeta, \mathcal{P}, \epsilon)$

---

1: $\epsilon_{cgs} \leftarrow \frac{\epsilon}{64\kappa}, \epsilon_{mp} \leftarrow 8\gamma^2 \left( \frac{4\kappa L \epsilon_{cgs}}{\alpha^2} + \frac{2\zeta}{\alpha} + \frac{2\sigma^2}{|\mathcal{P}|\alpha^2} \right), R \leftarrow \left\lceil \log_2 \frac{4D_\mathcal{X}^2}{\epsilon_{mp}} \right\rceil$.
2: **for** $r = 1, \ldots, R$ **do**
3:      Use iSTORC method (Algorithm 5) with objective function $-f(\mathbf{x}_{r-1}, \cdot)$ and start point $\mathbf{y}_0$ to
     compute $\mathbf{y}_r$ such that: $\mathbb{E}[f(\mathbf{x}_{r-1}, \mathbf{y}_r)] \geq \mathbb{E}[\max_{\mathbf{y} \in \mathcal{Y}} f(\mathbf{x}_{r-1}, \mathbf{y})] - \epsilon_{cgs}$
4:      $\mathbf{v}_r \leftarrow \text{CndG}(\nabla_{\mathbf{x}} f_{\mathcal{P}}(\mathbf{z}, \mathbf{y}_r), \mathbf{v}, \alpha, \zeta, \mathcal{X}), \quad \mathbf{x}_r \leftarrow (1 - \gamma)\mathbf{x}_0 + \gamma \mathbf{v}_r$
5: **end for**

---

sets from the LIBSVM repository[2]: rcv1 ($n = 15,564, d = 47,236, h = 53$), sector ($n = 6,412, d = 55,197, h = 105$) and news20 ($n = 15,935, d = 62,061, h = 20$).

Since the primal-dual gap is hard to compute, we evaluate algorithms by the following FW-gap [13]:

$$\mathcal{G}(\mathbf{x}, \mathbf{y}) = \max_{\mathbf{u} \in \mathcal{X}} \langle \mathbf{x} - \mathbf{u}, \nabla_{\mathbf{x}} f(\mathbf{x}, \mathbf{y}) \rangle + \max_{\mathbf{v} \in \mathcal{Y}} \langle \mathbf{y} - \mathbf{v}, -\nabla_{\mathbf{y}} f(\mathbf{x}, \mathbf{y}) \rangle.$$

which is an upper bound of primal-dual gap and easy to compute. We measure the actual running time rather than number of iterations because the computational cost of projection, linear optimization and computing gradients are quite different.

We implement the mini-batch version of SVRE with batch size 100. The learning rate of SVRE is searched in $\{10^{-1}, 10^{-2}, \ldots, 10^{-6}\}$. On the other hand, the parameters of projection-free methods follows what the theory suggests. We report the experimental result in Figure 1.

In all experiments, our methods outperform baselines. The SVRE only performs a few iterations due to its heavy computational cost of the projection on to the trace norm ball. SPFW converges slowly for it does not have theoretical guarantee on the convex-strongly-concave case. We also find that MPSCGS converges faster than MPCGS, because the stochastic algorithms take advantages when $n$ is very large.

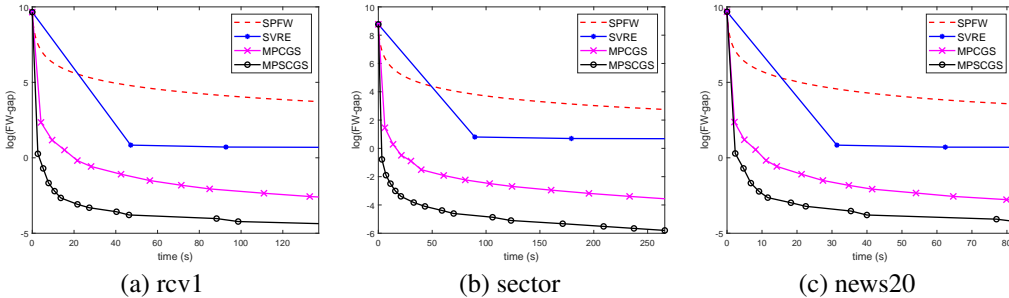

(a) rcv1            (b) sector            (c) news20

Figure 1: We demonstrate the perfomance of algorithms by time (s) versus $\log$(FW-gap) for robust multiclass classification with nuclear norm ball constraint on datasets "rcv1", "sector", and "news20".

# 6 Conclusion and Future Works

In this paper, we propose projection-free algorithms for solving saddle point problems with complicated constraints in both batch and stochastic settings. Our methods are purely projection-free and do not require that the saddle point problem has special structures. We also provide convergence analysis for our algorithms in the convex-strongly-concave case. The experimental results demonstrate the effectiveness of our algorithms on three real world data sets. On the other hand, we believe that there is room for improving the complexity of the LO oracles, which will be our future studies. In addition, we will investigate how to extend our framework to the general convex-concave case and establish stronger convergence results in the strongly-convex-strongly-concave case.

## Broader Impact

This paper studies projection-free algorithms for convex-strongly-concave saddle point problems. From a theoretical viewpoint, our method propose the first stochastic projection-free algorithm for saddle point problems without special conditions on the problem. From a practical viewpoint, our method can be applied to many machine learning applications which solve minimax problem with complicated constraints, e.g. robust optimization, matrix completion, two-player games and much more.

## Acknowledgments and Disclosure of Funding

The team is supported by "New Generation of AI 2030" Major Project (2018AAA0100900) and National Natural Science Foundation of China (61702327, 61772333, 61632017). Luo Luo is supported by GRF 16201320.

## Footnotes

[2]https://www.csie.ntu.edu.tw/ cjlin/libsvmtools/datasets/

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
