[Supplementary Material]

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

# A Proofs For MPCGS

In this section, we assume $f(\mathbf{x}, \mathbf{y})$ satisfies Assumption 1.

## A.1 Definitions and Lemmas

We define the following functions:

$$\mathbf{y}^*(\mathbf{x}) = \arg\max_{\mathbf{y} \in \mathcal{Y}} f(\mathbf{x}, \mathbf{y}),$$

$$\psi_k(\mathbf{x}) = (1 - \gamma_k)\mathbf{x}_{k-1} + \gamma_k \cdot \arg\min_{\mathbf{v} \in \mathcal{X}} \left\{ \nabla_{\mathbf{x}} f(\mathbf{z}_k, \mathbf{y}^*(\mathbf{x}))^\top \mathbf{v} + \frac{\alpha_k}{2} \|\mathbf{v} - \mathbf{v}_{k-1}\|^2 \right\}.$$

Since $f(\mathbf{x}, \cdot)$ is $\mu$-strongly-concave, $\mathbf{y}^*(\mathbf{x})$ is unique. Then, we have the following two lemmas.

**Lemma 1** ([20, Lemma 4.3]). $\mathbf{y}^*(\mathbf{x})$ *is $\kappa$-Lipschitz continuous.*

**Lemma 2.** $\psi_k(\mathbf{x})$ *is a $\frac{1}{2}$-contraction.*

*Proof.* Let

$$\nabla_1 = \nabla_{\mathbf{x}} f(\mathbf{z}_k, \mathbf{y}^*(\mathbf{x}_1)), \quad \mathbf{v}_1 = \arg\min_{\mathbf{v} \in \mathcal{X}} \left\{ \nabla_1^\top \mathbf{v} + \frac{\alpha_k}{2} \|\mathbf{v} - \mathbf{v}_{k-1}\|^2 \right\};$$

$$\nabla_2 = \nabla_{\mathbf{x}} f(\mathbf{z}_k, \mathbf{y}^*(\mathbf{x}_2)), \quad \mathbf{v}_2 = \arg\min_{\mathbf{v} \in \mathcal{X}} \left\{ \nabla_2^\top \mathbf{v} + \frac{\alpha_k}{2} \|\mathbf{v} - \mathbf{v}_{k-1}\|^2 \right\}.$$

Then we have

$$\|\psi_k(\mathbf{x}_1) - \psi_k(\mathbf{x}_2)\| = \gamma_k \|\mathbf{v}_1 - \mathbf{v}_2\|.$$

According to the optimality of $\mathbf{v}_1$ and $\mathbf{v}_2$, we have

$$\langle \nabla_1 + \alpha_k(\mathbf{v}_1 - \mathbf{v}_{k-1}), \mathbf{v}_1 - \mathbf{v}_2 \rangle \le 0;$$

$$\langle \nabla_2 + \alpha_k(\mathbf{v}_2 - \mathbf{v}_{k-1}), \mathbf{v}_2 - \mathbf{v}_1 \rangle \le 0.$$

Summing over inequalities, we get

$$\langle \nabla_1 - \nabla_2, \mathbf{v}_1 - \mathbf{v}_2 \rangle + \alpha_k \|\mathbf{v}_1 - \mathbf{v}_2\|^2 \le 0$$

Thus,

$$\|\mathbf{v}_1 - \mathbf{v}_2\|^2 \le -\langle \nabla_1 - \nabla_2, \mathbf{v}_1 - \mathbf{v}_2 \rangle / \alpha_k \le \frac{1}{2} \|\mathbf{v}_1 - \mathbf{v}_2\|^2 + \frac{1}{2} \|\nabla_1 - \nabla_2\|^2 / \alpha_k^2,$$

which indicates that

$$\|\mathbf{v}_1 - \mathbf{v}_2\| \le \frac{\|\nabla_1 - \nabla_2\|}{\alpha_k} \overset{(a)}{\le} \frac{L\|\mathbf{y}^*(\mathbf{x}_1) - \mathbf{y}^*(\mathbf{x}_2)\|}{\alpha_k} \overset{(b)}{\le} \frac{\kappa L\|\mathbf{x}_1 - \mathbf{x}_2\|}{\alpha_k}$$

Here (a) follows from the $L$-smoothness of $f$ and (b) follows from Lemma 1.
Since $\alpha_k = \frac{6\kappa_y L}{k+1}$ and $\gamma_k = \frac{3}{k+2}$, we have

$$\|\psi_k(\mathbf{x}_1) - \psi_k(\mathbf{x}_2)\| \le \frac{\gamma_k \kappa L \|\mathbf{x}_1 - \mathbf{x}_2\|}{\alpha_k} \le \frac{1}{2} \|\mathbf{x}_1 - \mathbf{x}_2\|.$$

$\square$

**Lemma 3.** *Assume the input parameters of the procedure Prox-step (Algorithm 4) is choosed as follows:*

$$\gamma = \frac{3}{k+2}, \quad \alpha = \frac{6\kappa L}{k+1}, \quad \zeta = \frac{LD_{\mathcal{X}}^2}{384k(k+1)}, \quad \epsilon = \frac{\kappa L D_{\mathcal{X}}^2}{k(k+1)(k+2)}.$$

*Then the Prox-step returns $(\mathbf{x}_R, \mathbf{y}_R, \mathbf{v}_R)$ which satisfies*

$$f(\mathbf{x}_R, \mathbf{y}_R) \ge \max_{\mathbf{y} \in \mathcal{Y}} f(\mathbf{x}_R, \mathbf{y}) - \epsilon.$$

*Proof.* Let $\psi(\mathbf{x}) = (1-\gamma)\mathbf{x}_0 + \gamma \arg\min_{\mathbf{u}\in\mathcal{X}}\{\nabla_\mathbf{x} f(\mathbf{z},\mathbf{y}^*(\mathbf{x}_r))^\top \mathbf{u} + \frac{\alpha}{2}\|\mathbf{v}-\mathbf{u}\|^2\}$. According to Lemma 2, $\psi(\mathbf{x})$ is a $\frac{1}{2}$-contraction.

By the optimality, we get

$$\langle \nabla_\mathbf{x} f(\mathbf{z},\mathbf{y}^*(\mathbf{x}_{r-1})) + \alpha(\mathbf{v}_{r-1}^* - \mathbf{v}), \mathbf{v}_{r-1}^* - \mathbf{v}_r\rangle \le 0. \tag{6}$$

Since $\mathbf{v}_r = \mathrm{CndG}(\nabla_\mathbf{x} f(\mathbf{z},\mathbf{y}_r),\mathbf{v},\alpha,\zeta,\mathcal{X})$, we have

$$\langle \nabla_\mathbf{x} f(\mathbf{z},\mathbf{y}_r) + \alpha(\mathbf{v}_r - \mathbf{v}), \mathbf{v}_r - \mathbf{v}_{r-1}^*\rangle \le \zeta. \tag{7}$$

Sum Eq.(6) and Eq.(7) together, we have

$$\langle \nabla_\mathbf{x} f(\mathbf{z},\mathbf{y}^*(\mathbf{x}_{r-1})) - \nabla_\mathbf{x} f(\mathbf{z},\mathbf{y}_r), \mathbf{v}_{r-1}^* - \mathbf{v}_r\rangle + \alpha\|\mathbf{v}_{r-1}^* - \mathbf{v}_r\|^2 \le \zeta.$$

Thus, we can get

$$
\begin{aligned}
\|\mathbf{v}_{r-1}^* - \mathbf{v}_r\|^2 &\le \frac{\zeta}{\alpha} - \frac{1}{\alpha}\langle \nabla_\mathbf{x} f(\mathbf{z},\mathbf{y}^*(\mathbf{x}_{r-1})) - \nabla_\mathbf{x} f(\mathbf{z},\mathbf{y}_r), \mathbf{v}_{r-1}^* - \mathbf{v}_r\rangle \\
&\le \frac{\zeta}{\alpha} + \frac{1}{2}\|\mathbf{v}_{r-1}^* - \mathbf{v}_r\|^2 + \frac{\|\nabla_\mathbf{x} f(\mathbf{z},\mathbf{y}^*(\mathbf{x}_{r-1})) - \nabla_\mathbf{x} f(\mathbf{z},\mathbf{y}_r)\|^2}{2\alpha^2},
\end{aligned}
$$

which means

$$\|\mathbf{v}_{r-1}^* - \mathbf{v}_r\|^2 \le \frac{2\zeta}{\alpha} + \frac{\|\nabla_\mathbf{x} f(\mathbf{z},\mathbf{y}^*(\mathbf{x}_{r-1})) - \nabla_\mathbf{x} f(\mathbf{z},\mathbf{y}_r)\|^2}{\alpha^2}.$$

By the $L$-smoothness of $f$, we have

$$
\begin{aligned}
\|\mathbf{v}_{r-1}^* - \mathbf{v}_r\| &\le \sqrt{\frac{L^2}{\alpha^2}\|\mathbf{y}^*(\mathbf{x}_{r-1}) - \mathbf{y}_r\|^2 + \frac{2\zeta}{\alpha}} \\
&\overset{(a)}{\le} \sqrt{\frac{2\kappa L}{\alpha^2}(f(\mathbf{x}_{r-1},\mathbf{y}^*(x_{r-1})) - f(\mathbf{x}_{r-1},\mathbf{y}_r)) + \frac{2\zeta}{\alpha}} \\
&\overset{(b)}{\le} \sqrt{\frac{2\kappa L\epsilon_{cgs}}{\alpha^2} + \frac{2\zeta}{\alpha}}
\end{aligned}
\tag{8}
$$

where (a) is by the $\mu$-strongly-concavity of $f(\mathbf{x},\cdot)$ and (b) is by the stopping condition of CGS. Assume the fix point of $\psi(\cdot)$ is $\tilde{\mathbf{x}}$. Then we bound $\|\mathbf{x}_r - \tilde{\mathbf{x}}\|$ as follows:

$$
\begin{aligned}
\|\mathbf{x}_r - \tilde{\mathbf{x}}\| &= \|(1-\gamma)\mathbf{x}_{k-1} + \gamma\mathbf{v}_r - \psi(\tilde{\mathbf{x}})\| \\
&\le \|\psi(\mathbf{x}_{r-1}) - \psi(\tilde{\mathbf{x}})\| + \|(1-\gamma)\mathbf{x}_{k-1} + \gamma\mathbf{v}_r - (1-\gamma)\mathbf{x}_{k-1} - \gamma\mathbf{v}_{r-1}^*\| \\
&\le \frac{1}{2}\|\mathbf{x}_{r-1} - \tilde{\mathbf{x}}\| + \gamma\|\mathbf{v}_r - \mathbf{v}_{r-1}^*\| \\
&\le \frac{1}{2}\|\mathbf{x}_{r-1} - \tilde{\mathbf{x}}\| + \frac{\epsilon_{mp}}{4} \\
&\le 2^{-r}\|\mathbf{x}_0 - \tilde{\mathbf{x}}\| + \frac{\epsilon_{mp}}{2}
\end{aligned}
$$

where the second inequality follows from Lemma 2. Since $R = \left\lceil \log_2 \frac{4D_\mathcal{X}}{\epsilon_{mp}}\right\rceil$, we know that

$$\|\mathbf{x}_{R-1} - \tilde{\mathbf{x}}\| \le 2\cdot 2^{-R}\|\mathbf{x}_0 - \tilde{\mathbf{x}}\| + \frac{\epsilon_{mp}}{2} \le 2\cdot\frac{\epsilon_{mp}}{4} + \frac{\epsilon_{mp}}{2} = \epsilon_{mp}.$$

Then, we can get

$$
\begin{aligned}
\|\mathbf{y}_R - \mathbf{y}^*(\mathbf{x}_R)\| &\le \|\mathbf{y}^*(\mathbf{x}_R) - \mathbf{y}^*(\tilde{\mathbf{x}})\| + \|\mathbf{y}^*(\mathbf{x}_{R-1}) - \mathbf{y}^*(\tilde{\mathbf{x}})\| + \|\mathbf{y}^*(\mathbf{x}_{R-1}) - \mathbf{y}_R\| \\
&\le \kappa(\|\mathbf{x}_R - \tilde{\mathbf{x}}\| + \|\mathbf{x}_{R-1} - \tilde{\mathbf{x}}\|) + \sqrt{\frac{2}{\mu}\epsilon_{cgs}} \\
&\le 2\kappa\epsilon_{mp} + \sqrt{\frac{2}{\mu}\epsilon_{cgs}} \\
&= 8\kappa\gamma_k\sqrt{\frac{2\kappa L\epsilon_{cgs}}{\alpha_k^2} + \frac{2\zeta_k}{\alpha_k}} + \sqrt{\frac{2}{\mu}\epsilon_{cgs}}
\end{aligned}
$$

where the second inequality comes from Lemma 1 and the concavity of $f(\mathbf{x}, \cdot)$. According to the value of input parameters, we can get

$$\|\mathbf{y}_R - \mathbf{y}^*(\mathbf{x}_R)\| \leq \sqrt{\frac{\kappa D_\mathcal{X}^2}{k(k+1)(k+2)}} + \sqrt{\frac{\kappa D_\mathcal{X}^2}{32k(k+1)(k+2)}} \leq \sqrt{\frac{2\kappa D_\mathcal{X}^2}{k(k+1)(k+2)}}.$$

By the $L$-smoothness of $f$ and the optimality of $\mathbf{y}^*(\mathbf{x}_R)$, we have

$$\begin{aligned}
f(\mathbf{x}_R, \mathbf{y}_R) \geq &f(\mathbf{x}_R, \mathbf{y}^*(\mathbf{x}_R)) + \langle \nabla_\mathbf{y} f(\mathbf{x}_R, \mathbf{y}^*(\mathbf{x}_R)), \mathbf{y}_R - \mathbf{y}^*(\mathbf{x}_R) \rangle - \frac{L}{2}\|\mathbf{y}_R - \mathbf{y}^*(\mathbf{x}_R)\|^2 \\
\geq &f(\mathbf{x}_R, \mathbf{y}^*(\mathbf{x}_R)) - \frac{L}{2}\|\mathbf{y}_R - \mathbf{y}^*(\mathbf{x}_R)\|^2 \\
\geq &f(\mathbf{x}_R, \mathbf{y}^*(\mathbf{x}_R)) - \frac{\kappa L D_\mathcal{X}^2}{k(k+1)(k+2)} = f(\mathbf{x}_R, \mathbf{y}^*(\mathbf{x}_R)) - \epsilon.
\end{aligned}$$

$\square$

In addition, by the updating formula of Prox-step, it is obvious that the Line 4 of Algorithm 3 ensures that $\mathbf{v}_k = \text{CndG}(\nabla_\mathbf{x} f(\mathbf{z}_k, \mathbf{y}_k), \mathbf{v}_{k-1}, \alpha_k, \zeta_k, \mathcal{X})$ and $\mathbf{x}_k = (1 - \gamma_k)\mathbf{x}_{k-1} + \gamma_k \mathbf{v}_k$.

## A.2 Proof of Theorem 1

*Proof.* According to the smoothness, for any $\tilde{\mathbf{x}} \in \mathcal{X}$, we have

$$\begin{aligned}
&f(\mathbf{x}_k, \mathbf{y}_k) \\
\leq &f(\mathbf{z}_k, \mathbf{y}_k) + \nabla_\mathbf{x} f(\mathbf{z}_k, \mathbf{y}_k)^\top (\mathbf{x}_k - \mathbf{z}_k) + \frac{L}{2}\|\mathbf{x}_k - \mathbf{z}_k\|^2 \\
= &(1-\gamma_k)(f(\mathbf{z}_k, \mathbf{y}_k) + \nabla_\mathbf{x} f(\mathbf{z}_k, \mathbf{y}_k)^\top(\mathbf{x}_{k-1} - \mathbf{z}_k)) + \gamma_k(f(\mathbf{z}_k, \mathbf{y}_k) + \nabla_\mathbf{x} f(\mathbf{z}_k, \mathbf{y}_k)^\top(\tilde{\mathbf{x}} - \mathbf{z}_k)) \\
&+ \gamma_k \nabla_\mathbf{x} f(\mathbf{z}_k, \mathbf{y}_k)^\top(\mathbf{v}_k - \tilde{\mathbf{x}}) + \frac{\gamma_k^2 L}{2}\|\mathbf{v}_k - \mathbf{v}_{k-1}\|^2 \\
\leq &(1-\gamma_k)f(\mathbf{x}_{k-1}, \mathbf{y}_k) + \gamma_k f(\tilde{\mathbf{x}}, \mathbf{y}_k) + \gamma_k \nabla_\mathbf{x} f(\mathbf{z}_k, \mathbf{y}_k)^\top(\mathbf{v}_k - \tilde{\mathbf{x}}) + \frac{\gamma_k^2 L}{2}\|\mathbf{v}_k - \mathbf{v}_{k-1}\|^2.
\end{aligned} \quad (9)$$

where the last inequality comes from the convexity of $f(\cdot, \mathbf{y}_k)$. Notice that the update of $\mathbf{v}_k$ and the stopping condition of CndG procedure ensures that

$$\max_{\mathbf{x} \in \mathcal{X}} \langle \nabla_\mathbf{x} f(\mathbf{z}_k, \mathbf{y}_k) + \alpha_k(\mathbf{v}_k - \mathbf{v}_{k-1}), \mathbf{v}_k - \mathbf{x} \rangle \leq \zeta_k. \quad (10)$$

Combining Eq.(9) and (10) we can get

$$\begin{aligned}
&f(\mathbf{x}_k, \mathbf{y}_k) - f(\tilde{\mathbf{x}}, \mathbf{y}_k) \\
\leq &(1-\gamma_k)(f(\mathbf{x}_{k-1}, \mathbf{y}_k) - f(\tilde{\mathbf{x}}, \mathbf{y}_k)) + \gamma_k \zeta_k - \gamma_k \alpha_k(\mathbf{v}_k - \mathbf{v}_{k-1})^\top(\mathbf{v}_k - \tilde{\mathbf{x}}) + \frac{\gamma_k^2 L}{2}\|\mathbf{v}_k - \mathbf{v}_{k-1}\|^2 \\
= &(1-\gamma_k)(f(\mathbf{x}_{k-1}, \mathbf{y}_k) - f(\tilde{\mathbf{x}}, \mathbf{y}_k)) + \gamma_k \zeta_k + \frac{\gamma_k \alpha_k}{2}\left(\|\mathbf{v}_{k-1} - \tilde{\mathbf{x}}\|^2 - \|\mathbf{v}_k - \tilde{\mathbf{x}}\|^2\right) \\
&+ \frac{\gamma_k}{2}(L\gamma_k - \alpha_k)\|\mathbf{v}_k - \mathbf{v}_{k-1}\|^2.
\end{aligned} \quad (11)$$

Let $\Phi(k) = k(k+1)(k+2)(f(\mathbf{x}_k, \mathbf{y}_k) - f(\tilde{\mathbf{x}}, \mathbf{y}_k))$. According to line 4 of Algorithm 3, we have

$$f(\mathbf{x}_k, \mathbf{y}_k) \geq \max_{\mathbf{y} \in \mathcal{Y}} f(\mathbf{x}_k, \mathbf{y}) - \epsilon_k.$$

Thus, we have

$$
\begin{aligned}
\Phi(k) \leq & \Phi(k-1) + k(k-1)(k+1)(f(\mathbf{x}_{k-1}, \mathbf{y}_k) - f(\mathbf{x}_{k-1}, \mathbf{y}_{k-1}) - f(\tilde{\mathbf{x}}, \mathbf{y}_k) + f(\tilde{\mathbf{x}}, \mathbf{y}_{k-1})) \\
& + k(k+1)(k+2)\left(\gamma_k \eta_t + \frac{\gamma_k \alpha_k}{2}(\|\mathbf{v}_{k-1} - \tilde{\mathbf{x}}\|^2 - \|\mathbf{v}_k - \tilde{\mathbf{x}}\|^2)\right) \\
\leq & \Phi(k-1) + k(k-1)(k+1)(\epsilon_{k-1} - f(\tilde{\mathbf{x}}, \mathbf{y}_k) + f(\tilde{\mathbf{x}}, \mathbf{y}_{k-1})) \\
& + k(k+1)(k+2)\left(\gamma_k \zeta_k + \frac{\gamma_k \alpha_k}{2}(\|\mathbf{v}_{k-1} - \tilde{\mathbf{x}}\|^2 - \|\mathbf{v}_k - \tilde{\mathbf{x}}\|^2)\right) \\
\leq & \Phi(0) + \sum_{s=1}^{k} s(s-1)(s+1)\epsilon_{s-1} - \sum_{s=1}^{k} s(s-1)(s+1)(f(\tilde{\mathbf{x}}, \mathbf{y}_s) - f(\tilde{\mathbf{x}}, \mathbf{y}_{s-1})) \\
& + \sum_{s=1}^{k} s(s+1)(s+2)\left(\gamma_s \zeta_s + \frac{\gamma_s \alpha_s}{2}(\|\mathbf{v}_{s-1} - \tilde{\mathbf{x}}\|^2 - \|\mathbf{v}_s - \tilde{\mathbf{x}}\|^2)\right)
\end{aligned}
\tag{12}
$$

Notice that

$$
\sum_{s=1}^{k} s(s+1)(s+2)\gamma_s \zeta_s = \frac{1}{128} k L D_{\mathcal{X}}^2
\tag{13}
$$

and

$$
\begin{aligned}
& \sum_{s=1}^{k} s(s+1)(s+2)\frac{\gamma_s \alpha_s}{2}(\|\mathbf{v}_{s-1} - \tilde{\mathbf{x}}\|^2 - \|\mathbf{v}_s - \tilde{\mathbf{x}}\|^2) \\
= & 9\kappa L \sum_{s=1}^{k} s(\|\mathbf{v}_{s-1} - \tilde{\mathbf{x}}\|^2 - \|\mathbf{v}_s - \tilde{\mathbf{x}}\|^2) \\
\leq & 9\kappa L \sum_{s=0}^{k-1} \|\mathbf{v}_s - \tilde{\mathbf{x}}\|^2 \leq 9k\kappa L D_{\mathcal{X}}^2.
\end{aligned}
\tag{14}
$$

Substituting Eq.(13) and Eq.(14) into Eq.(12) and using the fact that $\Phi(0) = 0$, we have

$$
\Phi(k) \leq \sum_{s=1}^{k} s(s-1)(s+1)\epsilon_{s-1} - \sum_{s=1}^{k} s(s-1)(s+1)(f(\tilde{\mathbf{x}}, \mathbf{y}_s) - f(\tilde{\mathbf{x}}, \mathbf{y}_{s-1})) + 10k\kappa L D_{\mathcal{X}}^2.
$$

Thus, for any $\tilde{\mathbf{y}} \in \mathcal{Y}$, we have

$$
\begin{aligned}
& \sum_{s=1}^{k} s(s-1)(s+1)\epsilon_{s-1} + 10k\kappa L D_{\mathcal{X}}^2 \\
\geq & \Phi(k) + \sum_{s=1}^{k} s(s-1)(s+1)(f(\tilde{\mathbf{x}}, \mathbf{y}_s) - f(\tilde{\mathbf{x}}, \mathbf{y}_{s-1})) \\
\geq & \Phi(k) - 3\sum_{s=1}^{k-1} s(s+1)f(\tilde{\mathbf{x}}, \mathbf{y}_s) + k(k-1)(k+1)f(\tilde{\mathbf{x}}, \mathbf{y}_k) \\
= & k(k+1)(k+2)f(\mathbf{x}_k, \mathbf{y}_k) - 3\sum_{s=1}^{k} s(s+1)f(\tilde{\mathbf{x}}, \mathbf{y}_s) \\
\overset{(a)}{\geq} & k(k+1)(k+2)f[f(\mathbf{x}_k, \mathbf{y}_k) - f(\tilde{\mathbf{x}}, \bar{\mathbf{y}}_k)] \\
\geq & k(k+1)(k+2)[f(\mathbf{x}_k, \tilde{\mathbf{y}}) - f(\tilde{\mathbf{x}}, \bar{\mathbf{y}}_k) - \epsilon_k]
\end{aligned}
$$

where (a) is by the concavity of $f(\mathbf{x}, \cdot)$ and the definition of $\bar{\mathbf{y}}_k$. Then, we can obtain the bound of the primal-dual gap:

$$
f(\mathbf{x}_k, \tilde{\mathbf{y}}) - f(\tilde{\mathbf{x}}, \bar{\mathbf{y}}_k) \leq \frac{1}{k(k+1)(k+2)}\left(\sum_{s=1}^{k} s(s+1)(s+2)\epsilon_s + 10k\kappa L D_{\mathcal{X}}^2\right) \leq \frac{11\kappa L D_{\mathcal{X}}^2}{(k+1)(k+2)}
$$

where the last equation comes from the fact that $\epsilon_s = \frac{\kappa L D_{\mathcal{X}}^2}{s(s+1)(s+2)}$.

$\square$

## A.3   Proof of Corollary 1

According to Theorem 2.5 of [19], the CGS method requires $\mathcal{O}\left(\sqrt{\kappa}\log\frac{\kappa L D_{\mathcal{Y}}^2}{\epsilon_k}\right)$ FO calls and $\mathcal{O}\left(\frac{\kappa L D_{\mathcal{Y}}^2}{\epsilon_k}\right)$ LO calls, where $R$ is the number of iteration of the Prox-step procedure. Thus, the FO and LO complexity of MPCGS are respectively $\mathcal{O}\left(\sum_{k=1}^{N}\sum_{r=1}^{R}\sqrt{\kappa}\log\frac{\kappa L D_{\mathcal{Y}}^2}{\epsilon_k}\right)$ and $\mathcal{O}\left(\sum_{k=1}^{N}\sum_{r=1}^{R}\left(\frac{\kappa L D_{\mathcal{Y}}^2}{\epsilon_k}+\frac{\alpha_k D_{\mathcal{X}}^2}{\zeta_k}\right)\right)$. Theorem 1 implies that $N$ should be the order $\Theta\left(\sqrt{\frac{\kappa L D_{\mathcal{X}}^2}{\epsilon}}\right)$. Plugging in all parameters obtains the complexity of MPCGS.

# B   Proof of Inexact STORC Algorithm

In this section, we provide the details for the analysis of iSTORC (Algorithm 5). We suppose that the objective function $h(\mathbf{x})$ satisfies assumptions in Section 4.1. We define $D_t = \sqrt{\frac{\kappa D^2}{2^{t-1}}}$ for any $t \in [N]$.

## B.1   Technical Lemmas

**Lemma 4** (Lemma 3 of [11]). *At the $t$-th outer iteration of iSTORC (Algorithm 5), we suppose that $\mathbb{E}[\|\mathbf{x}_0 - \mathbf{x}^*\|] \leq D_t^2$. Then for any $k$, we have*

$$\mathbb{E}[h(\mathbf{x}_k) - h(\mathbf{x}^*)] \leq \frac{8LD_t^2}{k(k+1)}$$

*if $\mathbb{E}[\|\mathbf{r}_s - \nabla h(\mathbf{w}_s)\|^2] \leq \frac{L^2 D_t^2}{Ms(s+1)}$ for all $s \leq k$.*

**Lemma 5.** *For iSTORC, we have*

$$\mathbb{E}[\|\nabla h_{\mathcal{Q}_t}(\mathbf{x}) - \nabla h(\mathbf{x})\|^2] \leq \frac{L^2 D_t^2}{4M^2(M+1)}.$$

*Proof.* In expectation setting, we have

$$\mathbb{E}[\|\nabla h_{\mathcal{Q}_t}(\mathbf{x}) - \nabla h(\mathbf{x})\|^2] \leq \frac{\sigma^2}{Q_t} \leq \frac{L^2 D_t^2}{4M^2(M+1)}.$$

$\square$

## B.2   Proof of Theorem 2

*Proof.* We prove by this theorem by induction. For $t = 0$, by the smoothness and the convexity of $h$, we have

$$h(\bar{\mathbf{x}}_0) \leq h(\mathbf{x}^*) + \nabla h(\mathbf{x}^*)^\top(\bar{\mathbf{x}}_0 - \mathbf{x}^*) + \frac{L}{2}\|\bar{\mathbf{x}}_0 - \mathbf{x}^*\|^2 \leq h(\mathbf{x}^*) + \frac{LD^2}{2}.$$

Then, we suppose $\mathbb{E}[h(\bar{\mathbf{x}}_{t-1}) - h(\mathbf{x}^*)] \leq \frac{LD^2}{2^t}$ and consider the $t$-th outer iteration. By the strong convexity and the inductive assumption, we have

$$\mathbb{E}[\|\mathbf{x}_0 - \mathbf{x}^*\|^2] \leq \frac{2}{\mu}\mathbb{E}[h(\bar{\mathbf{x}}_{t-1}) - h(\mathbf{x}^*)] \leq \frac{LD^2}{\mu 2^{t-1}} = D_t^2.$$

Now we use another induction on the inner iteration to show that it holds $\mathbb{E}[h(\mathbf{x}_k) - h(\mathbf{x}^*)] \leq \frac{8LD_t^2}{k(k+1)}$ for any $1 \leq k \leq M$.

For $k = 1$, by the fact that $\mathbf{w}_1 = \mathbf{x}_0$ and Lemma 5, we have

$$\mathbb{E}[\|\mathbf{r}_1 - \nabla h(\mathbf{w}_1)\|^2] = \mathbb{E}[\|\nabla h_{\mathcal{Q}_t}(\mathbf{x}_0) - \nabla h(\mathbf{x}_0)\|^2] \leq \frac{L^2 D_t^2}{4M}$$

for both finite-sum setting and expectation setting. Then, by Lemma 4 we can obtain

$$\mathbb{E}[h(\mathbf{x}_1) - h(\mathbf{x}^*)] \leq 4LD_t^2.$$

Now we suppose $\mathbb{E}[h(\mathbf{x}_s) - h(\mathbf{x}^*)] \leq \frac{8LD_t^2}{s(s+1)}$ for $s < k$ where $k \geq 2$. We consider the case $s = k$.

Since the variance of a variable is less than its second-order moment, we can get

$$\mathbb{E}_{\mathcal{S}_{t,s}, \mathcal{Q}_t}[\|\mathbf{r}_s - \nabla h(\mathbf{w}_s)\|^2]$$
$$=\mathbb{E}_{\mathcal{S}_{t,s}, \mathcal{Q}_t}[\|\nabla h_{\mathcal{S}_{t,s}}(\mathbf{w}_s) - \nabla h_{\mathcal{S}_{t,s}}(\mathbf{x}_0) + \nabla h_{\mathcal{Q}_t}(\mathbf{x}_0) - \nabla h(\mathbf{w}_s)\|^2]$$
$$\leq 2\mathbb{E}_{\mathcal{S}_{t,s}}[\|\nabla h_{\mathcal{S}_{t,s}}(\mathbf{w}_s) - \nabla h_{\mathcal{S}_{t,s}}(\mathbf{x}_0) - (\nabla h(\mathbf{w}_s) - \nabla h(\mathbf{x}_0))\|^2] + 2\mathbb{E}_{\mathcal{Q}_t}[\|\nabla h_{\mathcal{Q}_t}(\mathbf{x}_0) - \nabla h(\mathbf{x}_0)\|^2]$$
$$\leq 2\mathbb{E}_{\mathcal{S}_{t,s}}[\|\nabla h_{\mathcal{S}_{t,s}}(\mathbf{w}_s) - \nabla h_{\mathcal{S}_{t,s}}(\mathbf{x}_0)\|^2] + 2\mathbb{E}_{\mathcal{Q}_t}[\|\nabla h_{\mathcal{Q}_t}(\mathbf{x}_0) - \nabla h(\mathbf{x}_0)\|^2]$$
$$\leq \frac{2L^2}{S}\|\mathbf{w}_s - \mathbf{x}_0\|^2 + 2\mathbb{E}_{\mathcal{Q}_t}[\|\nabla h_{\mathcal{Q}_t}(\mathbf{x}_0) - \nabla h(\mathbf{x}_0)\|^2].$$

$$(15)$$

Since $\|\mathbf{w}_s - \mathbf{x}_0\|^2 \leq 2\|\mathbf{w}_s - \mathbf{x}^*\|^2 + 2\|\mathbf{x}_0 - \mathbf{x}^*\|^2$, then we bound $\mathbb{E}[\|\mathbf{x}_0 - \mathbf{x}^*\|^2]$ and $\mathbb{E}[\|\mathbf{w}_s - \mathbf{x}^*\|^2]$ separately.

For $\mathbb{E}[\|\mathbf{x}_0 - \mathbf{x}^*\|^2]$, we have

$$\mathbb{E}[\|\mathbf{x}_0 - \mathbf{x}^*\|^2] \leq D_t^2 = \frac{2\kappa D_t^2}{2\kappa} \leq \frac{64\kappa D_t^2}{M(M+1)}. \tag{16}$$

Since $\mathbf{w}_s = (1 - \lambda_s)\mathbf{x}_{s-1} + \lambda_s \mathbf{u}_{s-1}$ and $\mathbf{x}_{s-1} = (1 - \lambda_{s-1})\mathbf{x}_{s-2} + \lambda_{s-1}\mathbf{u}_{s-1}$, we have

$$\mathbf{w}_s = \frac{\lambda_{s-1} + \lambda_s - \lambda_{s-1}\lambda_s}{\lambda_{s-1}}\mathbf{x}_{s-1} - \frac{\lambda_s - \lambda_{s-1}\lambda_s}{\lambda_{s-1}}\mathbf{x}_{s-2}$$

Thus, we have

$$\|\mathbf{w}_s - \mathbf{x}^*\|^2 = \left\|\frac{\lambda_{s-1} + \lambda_s - \lambda_{s-1}\lambda_s}{\lambda_{s-1}}(\mathbf{x}_{s-1} - \mathbf{x}^*) - \frac{\lambda_s - \lambda_{s-1}\lambda_s}{\lambda_{s-1}}(\mathbf{x}_{s-2} - \mathbf{x}^*)\right\|^2$$
$$\leq 2\left\|\frac{\lambda_{s-1} + \lambda_s - \lambda_{s-1}\lambda_s}{\lambda_{s-1}}(\mathbf{x}_{s-1} - \mathbf{x}^*)\right\|^2 + 2\left\|\frac{\lambda_s - \lambda_{s-1}\lambda_s}{\lambda_{s-1}}(\mathbf{x}_{s-2} - \mathbf{x}^*)\right\|^2 \quad (17)$$
$$\leq 8\|\mathbf{x}_{s-1} - \mathbf{x}^*\|^2 + 2\|\mathbf{x}_{s-2} - \mathbf{x}^*\|^2$$

where the last inequality comes from the fact $\lambda_s \leq \lambda_{s-1} \leq 1$.

If $s = 2$, we can obtain

$$L^2 \mathbb{E}[\|\mathbf{w}_2 - \mathbf{x}_0\|^2] \leq L^2 \mathbb{E}[16\|\mathbf{x}_1 - \mathbf{x}^*\|^2 + 6\|\mathbf{x}_0 - \mathbf{x}^*\|^2]$$
$$\leq 32\kappa L \mathbb{E}[h(\mathbf{x}_1) - h(\mathbf{x}^*)] + \frac{384\kappa L^2 D_t^2}{(M+1)^2}$$
$$\leq \frac{768\kappa L^2 D_t^2}{s(s+1)} + \frac{384\kappa L^2 D_t^2}{s(s+1)} = \frac{1152\kappa L^2 D_t^2}{s(s+1)}$$

where the second inequality comes from the strong convexity of $h(\mathbf{x})$ and Eq. (16), the third inequality comes from the induction hypothesis.

If $s \geq 3$, we can obtain

$$L^2 \mathbb{E}[\|\mathbf{w}_s - \mathbf{x}^*\|^2] \leq 8L^2 \mathbb{E}[\|\mathbf{x}_{s-1} - \mathbf{x}^*\|^2] + 2L^2 \mathbb{E}[\|\mathbf{x}_{s-2} - \mathbf{x}^*\|^2]$$
$$\leq 16\kappa L \mathbb{E}[h(\mathbf{x}_{s-1}) - h(\mathbf{x}^*)] + 4\kappa L \mathbb{E}[h(\mathbf{x}_{s-2}) - h(\mathbf{x}^*)]$$
$$\leq \frac{128\kappa L^2 D_t^2}{s(s-1)} + \frac{32\kappa L^2 D_t^2}{(s-2)(s-1)} \leq \frac{448\kappa L^2 D_t^2}{s(s+1)}$$

$$(18)$$

which indicates that

$$L^2\mathbb{E}[\|\mathbf{w}_s - \mathbf{x}_0\|^2] \leq L^2\mathbb{E}[2\|\mathbf{w}_s - \mathbf{x}^*\|^2 + 2\|\mathbf{x}_0 - \mathbf{x}^*\|^2] \leq \frac{1024\kappa L^2 D_t^2}{s(s+1)}$$

where the last inequality is due to Eq.(16) and Eq.(18). Substituting all these pieces into Eq.(15), we can obtain

$$\begin{aligned}
\mathbb{E}_{\mathcal{S}_{t,s},\mathcal{Q}_t}[\|\mathbf{r}_s - \nabla h(\mathbf{w}_s)\|^2] &\leq \frac{2L^2}{S}\|\mathbf{w}_s - \mathbf{x}_0\|^2 + 2\mathbb{E}_{\mathcal{Q}_t}[\|\nabla h_{\mathcal{Q}_t}(\mathbf{x}_0) - \nabla h(\mathbf{x}_0)\|^2] \\
&\leq \frac{2400\kappa L^2 D_t^2}{s(s+1)S} + \frac{2\sigma^2}{Q_t} \\
&\leq \frac{L^2 D_t^2}{2Ms(s+1)} + \frac{L^2 D_t^2}{2Ms(s+1)} = \frac{L^2 D_t^2}{Ms(s+1)}
\end{aligned}$$

where the last inequality comes from the fact that $s \leq M$. By Lemma 4, we know that $\mathbb{E}[h(\mathbf{x}_s) - h(\mathbf{x}^*)] \leq \frac{8LD_t^2}{s(s+1)}$, which completes that induction. $\square$

### B.3 Proof of Corollary 2

*Proof.* Theorem 2 implies that $N$ should be the order $\Theta\left(\log_2 \frac{LD^2}{\epsilon}\right)$. Thus, the SFO complexity of iSTORC is

$$\mathcal{O}\left(\sum_{t=1}^{N}\left(Q_t + \sum_{k=1}^{M} S\right)\right) = \mathcal{O}\left(\left(\frac{\sqrt{\kappa}}{L\epsilon} + \kappa^2\right)\log_2\left(\frac{LD^2}{\epsilon}\right)\right).$$

Since the CndG procedure requires $\mathcal{O}\left(\frac{\beta_k D^2}{\eta_{t,k}}\right)$ iterations, the LO complexity of iSTORC is $\mathcal{O}\left(\frac{LD^2}{\epsilon}\right)$. $\square$

## C   Proof for MPSCGS

In this section, we assume $f(\mathbf{x}, \mathbf{y})$ satisfies Assumption 1 and Assumption 2. We also use the same definition of $\mathbf{y}^*(\cdot)$ and $\psi_k(\cdot)$ as in Section A.1. We denote $\sigma_k^2 = \frac{\sigma^2}{P_k}$ additionally.

### C.1   Notation and Lemma

**Lemma 6.** *We set the input of Stochastic-Prox-step (Algorithm 7) as*

$$\gamma = \frac{3}{k+2}, \quad \alpha = \frac{6\kappa L}{k+1}, \quad \zeta = \frac{LD_{\mathcal{X}}^2}{576k(k+1)}, \quad \epsilon = \frac{\kappa LD_{\mathcal{X}}^2}{k(k+1)(k+2)}, \quad P = \left\lceil\frac{96\sigma^2(k+1)^3}{\kappa L^2 D_{\mathcal{X}}^2}\right\rceil,$$

*then the output $(\mathbf{x}_R, \mathbf{y}_R, \mathbf{v}_R)$ satisfies*

$$\mathbb{E}[f(\mathbf{x}_R, \mathbf{y}_R)] \geq \mathbb{E}\left[\max_{\mathbf{y}\in\mathcal{Y}} f(\mathbf{x}_R, \mathbf{y})\right] - \epsilon.$$

*Proof.* Let $\psi(\mathbf{x}) = (1-\gamma)\mathbf{x}_0 + \gamma\arg\min_{\mathbf{u}\in\mathcal{X}}\left\{\nabla_{\mathbf{x}} f(\mathbf{z}, \mathbf{y}^*(\mathbf{x}_r))^\top \mathbf{u} + \frac{\alpha}{2}\|\mathbf{v} - \mathbf{u}\|^2\right\}$. According to Lemma 2, $\psi(\mathbf{x})$ is a $\frac{1}{2}$-contraction. Let $\mathbf{v}_r^* = \arg\min_{\mathbf{u}\in\mathcal{X}}\{\nabla_{\mathbf{x}} f(\mathbf{z}, \mathbf{y}^*(\mathbf{x}_r))^\top \mathbf{u} + \frac{\alpha}{2}\|\mathbf{v} - \mathbf{u}\|^2\}$, then

$$\psi(\mathbf{x}_r) = (1-\gamma)\mathbf{x}_0 + \gamma\mathbf{v}_r^*.$$

By the optimality, we get

$$\langle\nabla_{\mathbf{x}} f(\mathbf{z}, \mathbf{y}^*(\mathbf{x}_{r-1})) + \alpha(\mathbf{v}_{r-1}^* - \mathbf{v}), \mathbf{v}_{r-1}^* - \mathbf{v}_r\rangle \leq 0. \tag{19}$$

Since $\mathbf{v}_r = \text{CndG}(\nabla_{\mathbf{x}} f_{\mathcal{P}}(\mathbf{z}, \mathbf{y}_r), \mathbf{v}, \alpha, \zeta, \mathcal{X})$, we have

$$\langle\nabla_{\mathbf{x}} f(\mathbf{z}, \mathbf{y}_r) + \alpha(\mathbf{v}_r - \mathbf{v}), \mathbf{v}_r - \mathbf{v}_{r-1}^*\rangle \leq \zeta \tag{20}$$

Sum Eq.(19) and Eq.(20) together, we have

$$\langle \nabla_{\mathbf{x}} f(\mathbf{z}, \mathbf{y}^*(\mathbf{x}_{r-1})) - \nabla_{\mathbf{x}} f_{\mathcal{P}}(\mathbf{z}, \mathbf{y}_r), \mathbf{v}_{r-1}^* - \mathbf{v}_r \rangle + \alpha \|\mathbf{v}_{r-1}^* - \mathbf{v}_r\|^2 \leq \zeta$$

Thus, we can get

$$\mathbb{E}[\|\mathbf{v}_{r-1}^* - \mathbf{v}_r\|^2] \leq \frac{\zeta}{\alpha} - \frac{1}{\alpha} \mathbb{E}[\langle \nabla_{\mathbf{x}} f(\mathbf{z}, \mathbf{y}^*(\mathbf{x}_{r-1})) - \nabla_{\mathbf{x}} f_{\mathcal{P}}(\mathbf{z}, \mathbf{y}_r), \mathbf{v}_{r-1}^* - \mathbf{v}_r \rangle]$$

$$\leq \frac{\zeta}{\alpha} + \frac{1}{2} \mathbb{E}[\|\mathbf{v}_{r-1}^* - \mathbf{v}_r\|^2] + \frac{\mathbb{E}[\|\nabla_{\mathbf{x}} f(\mathbf{z}, \mathbf{y}^*(\mathbf{x}_{r-1})) - \nabla_{\mathbf{x}} f_{\mathcal{P}}(\mathbf{z}, \mathbf{y}_r)\|^2]}{2\alpha^2}$$

which indicates

$$\mathbb{E}[\|\mathbf{v}_{r-1}^* - \mathbf{v}_r\|^2]$$

$$\leq \frac{\mathbb{E}[\|\nabla_{\mathbf{x}} f(\mathbf{z}, \mathbf{y}^*(\mathbf{x}_{r-1})) - \nabla_{\mathbf{x}} f_{\mathcal{P}}(\mathbf{z}, \mathbf{y}_r)\|^2]}{\alpha^2} + \frac{2\zeta}{\alpha}$$

$$\leq \frac{\mathbb{E}[2\|\nabla_{\mathbf{x}} f(\mathbf{z}, \mathbf{y}^*(\mathbf{x}_{r-1})) - \nabla_{\mathbf{x}} f(\mathbf{z}, \mathbf{y}_r)\|^2 + 2\|\nabla_{\mathbf{x}} f(\mathbf{z}, \mathbf{y}_r) - \nabla_{\mathbf{x}} f_{\mathcal{P}}(\mathbf{z}, \mathbf{y}_r)\|^2]}{\alpha^2} + \frac{2\zeta}{\alpha}$$

Denote $\hat{\sigma}^2 = \frac{\sigma^2}{P}$. By the $L$-smoothness of $f$, we can obtain

$$\mathbb{E}[\|\mathbf{v}_{r-1}^* - \mathbf{v}_r\|^2] \leq \frac{2L^2}{\alpha^2} \mathbb{E}[\|\mathbf{y}^*(\mathbf{x}_{r-1}) - \mathbf{y}_r\|^2] + \frac{2\zeta}{\alpha} + \frac{2\hat{\sigma}^2}{\alpha^2}$$

$$\leq \frac{4\kappa L}{\alpha^2} \mathbb{E}[f(\mathbf{x}_{r-1}, \mathbf{y}^*(x_{r-1})) - f_{\mathcal{P}}(\mathbf{x}_{r-1}, \mathbf{y}_r)] + \frac{2\zeta}{\alpha} + \frac{2\hat{\sigma}^2}{\alpha^2}$$

$$\leq \frac{4\kappa L \epsilon_{cgs}}{\alpha^2} + \frac{2\zeta}{\alpha} + \frac{2\hat{\sigma}^2}{\alpha^2}$$

where we use the $\mu$-strongly-concavity of $f(\mathbf{x}, \cdot)$ and the stopping condition of iSTORC. Then we bound $\mathbb{E}[\|\mathbf{x}_r - \tilde{\mathbf{x}}\|^2]$ as follows:

$$\mathbb{E}[\|\mathbf{x}_r - \tilde{\mathbf{x}}\|^2] = \mathbb{E}[\|(1 - \gamma)\mathbf{z} + \gamma \mathbf{v}_r - \psi(\tilde{\mathbf{x}})\|^2]$$

$$\leq 2\mathbb{E}[\|\psi(\mathbf{x}_{r-1}) - \psi(\tilde{\mathbf{x}})\|^2] + 2\mathbb{E}[\|(1-\gamma)\mathbf{z} + \gamma \mathbf{v}_r - (1-\gamma)\mathbf{z} - \gamma \mathbf{v}_{r-1}^*\|^2]$$

$$\leq \frac{1}{2} \mathbb{E}[\|\mathbf{x}_{r-1} - \tilde{\mathbf{x}}\|^2] + 2\gamma^2 \mathbb{E}[\|\mathbf{v}_r - \mathbf{v}_{r-1}^*\|^2]$$

$$\leq \frac{1}{2} \mathbb{E}[\|\mathbf{x}_{r-1} - \tilde{\mathbf{x}}\|^2] + \frac{\epsilon_{mp}}{4}$$

$$\leq 2^{-r} \mathbb{E}[\|\mathbf{x}_0 - \tilde{\mathbf{x}}\|^2] + \frac{\epsilon_{mp}}{2}$$

Since $R = \left\lceil \log_2 \left( \frac{4D_{\mathcal{X}}^2}{\epsilon_{mp}} \right) \right\rceil$, we know that

$$\mathbb{E}[\|\mathbf{x}_{R-1} - \tilde{\mathbf{x}}\|^2] \leq 2 \cdot 2^{-R} \mathbb{E}[\|\mathbf{x}_0 - \tilde{\mathbf{x}}\|^2] + \frac{\epsilon_{mp}}{2} \leq 2 \cdot \frac{\epsilon_{mp}}{4} + \frac{\epsilon_{mp}}{2} = \epsilon_{mp}$$

Then, we can get

$$\mathbb{E}[\|\mathbf{y}_R - \mathbf{y}^*(\mathbf{x}_R)\|^2] \leq 3\mathbb{E}\left[\|\mathbf{y}^*(\mathbf{x}_R) - \mathbf{y}^*(\tilde{\mathbf{x}})\|^2 + \|\mathbf{y}^*(\mathbf{x}_{R-1}) - \mathbf{y}^*(\tilde{\mathbf{x}})\|^2 + \|\mathbf{y}^*(\mathbf{x}_{R-1}) - \mathbf{y}_R\|^2\right]$$

$$\leq 3\kappa \left(\mathbb{E}\left[\|\mathbf{x}_R - \tilde{\mathbf{x}}\|^2 + \|\mathbf{x}_{R-1} - \tilde{\mathbf{x}}\|^2\right]\right) + \frac{6}{\mu}\epsilon_{cgs}$$

$$\leq 6\kappa\epsilon_{mp} + \frac{6}{\mu}\epsilon_{cgs}$$

$$= 48\kappa\gamma^2 \left(\frac{4\kappa L\epsilon_{cgs}}{\alpha^2} + \frac{2\zeta}{\alpha} + \frac{2\sigma^2}{\alpha^2}\right) + \frac{6}{\mu}\epsilon_{cgs}$$

where the second inequality comes from Lemma 1 and the concavity of $f(\mathbf{x}, \cdot)$. According to the value of input parameters, we can get

$$\mathbb{E}[\|\mathbf{y}_R - \mathbf{y}^*(\mathbf{x}_R)\|^2] \leq \frac{2\kappa D_{\mathcal{X}}^2}{k(k+1)(k+2)}.$$

By the $L$-smoothness of $f$ and the optimality of $\mathbf{y}^*(\mathbf{x}_R)$, we have

$$
\begin{aligned}
\mathbb{E}[f(\mathbf{x}_R, \mathbf{y}_R)] &\geq \mathbb{E}[f(\mathbf{x}_R, \mathbf{y}^*(\mathbf{x}_R)) + \langle \nabla_\mathbf{y} f(\mathbf{x}_R, \mathbf{y}^*(\mathbf{x}_R)), \mathbf{y}_R - \mathbf{y}^*(\mathbf{x}_R) \rangle - \frac{L}{2}\|\mathbf{y}_R - \mathbf{y}^*(\mathbf{x}_R)\|^2] \\
&\geq \mathbb{E}[f(\mathbf{x}_R, \mathbf{y}^*(\mathbf{x}_R)) - \frac{L}{2}\|\mathbf{y}_R - \mathbf{y}^*(\mathbf{x}_R)\|^2] \\
&\geq \mathbb{E}\left[f(\mathbf{x}_R, \mathbf{y}^*(\mathbf{x}_R)) - \frac{\kappa L D_\mathcal{X}^2}{k(k+1)(k+2)}\right] = \mathbb{E}[f(\mathbf{x}_R, \mathbf{y}^*(\mathbf{x}_R))] - \epsilon.
\end{aligned}
$$

$\square$

## C.2 Proof of Theorem 3

*Proof.* Similar to Eq.(9) in section A.2, for any $\tilde{\mathbf{x}} \in \mathcal{X}$, we have

$$
f(\mathbf{x}_k, \mathbf{y}_k) \leq (1-\gamma_k)f(\mathbf{x}_{k-1}, \mathbf{y}_k) + \gamma_k f(\tilde{\mathbf{x}}, \mathbf{y}_k) + \gamma_k \nabla_\mathbf{x} f(\mathbf{z}_k, \mathbf{y}_k)^\top(\mathbf{v}_k - \tilde{\mathbf{x}}) + \frac{\gamma_k^2 L}{2}\|\mathbf{v}_k - \mathbf{v}_{k-1}\|^2.
$$

Let $\mathbf{d}_k = \nabla_\mathbf{x} f_{\mathcal{P}_k}(\mathbf{z}_k, \mathbf{y}_k)$. Notice that the update of $\mathbf{v}_k$ and the stopping condition of CndG procedure ensures that

$$
\max_{\mathbf{x} \in \mathcal{X}} \langle \mathbf{d}_k + \alpha_k(\mathbf{v}_k - \mathbf{v}_{k-1}), \mathbf{v}_k - \mathbf{x} \rangle \leq \zeta_k.
$$

Combining the previous two inequality, we can obtain

$$
\begin{aligned}
f(\mathbf{x}_k, \mathbf{y}_k) \leq & (1-\gamma_k)f(\mathbf{x}_{k-1}, \mathbf{y}_k) + \gamma_k f(\tilde{\mathbf{x}}, \mathbf{y}_k) + \gamma_k \mathbf{d}_k^\top(\mathbf{v}_k - \tilde{\mathbf{x}}) + \frac{\gamma_k^2 L}{2}\|\mathbf{v}_k - \mathbf{v}_{k-1}\|^2 \\
& + \gamma_k(\nabla_\mathbf{x} f(\mathbf{z}_k, \mathbf{y}_k) - \mathbf{d}_k)^\top(\mathbf{v}_k - \tilde{\mathbf{x}}) \\
\leq & (1-\gamma_k)f(\mathbf{x}_{k-1}, \mathbf{y}_k) + \gamma_k f(\tilde{\mathbf{x}}, \mathbf{y}_k) + \gamma_k \zeta_k - \gamma_k \alpha_k(\mathbf{v}_k - \mathbf{v}_{k-1})^\top(\mathbf{v}_k - \tilde{\mathbf{x}}) \\
& + \frac{\gamma_k^2 L}{2}\|\mathbf{v}_k - \mathbf{v}_{k-1}\|^2 + \gamma_k(\nabla_\mathbf{x} f(\mathbf{z}_k, \mathbf{y}_k) - \mathbf{d}_k)^\top(\mathbf{v}_k - \tilde{\mathbf{x}}) \\
= & (1-\gamma_k)f(\mathbf{x}_{k-1}, \mathbf{y}_k) + \gamma_k f(\tilde{\mathbf{x}}, \mathbf{y}_k) + \gamma_k \zeta_k + \frac{\gamma_k \alpha_k}{2}(\|\mathbf{v}_{k-1} - \tilde{\mathbf{x}}\|^2 - \|\mathbf{v}_k - \tilde{\mathbf{x}}\|^2) \\
& + \frac{\gamma_k}{2}\left((L\gamma_k - \alpha_k)\|\mathbf{v}_k - \mathbf{v}_{k-1}\|^2 + 2(\mathbf{d}_k - \nabla_\mathbf{x} f(\mathbf{z}_k, \mathbf{y}_k))^\top(\mathbf{v}_{k-1} - \mathbf{v}_k)\right) \\
& + \gamma_k(\mathbf{d}_k - \nabla_\mathbf{x} f(\mathbf{z}_k, \mathbf{y}_k))^\top(\tilde{\mathbf{x}} - \mathbf{v}_{k-1})
\end{aligned}
$$
(21)

Due to the fact $\alpha_k \geq L\gamma_k$ and $-\|\mathbf{a}\|^2 + 2\mathbf{a}^\top\mathbf{b} \leq \|\mathbf{b}\|^2$, we have

$$
(L\gamma_k - \alpha_k)\|\mathbf{v}_k - \mathbf{v}_{k-1}\|^2 + 2(\mathbf{d}_k - \nabla_\mathbf{x} f(\mathbf{z}_k, \mathbf{y}_k))^\top(\mathbf{v}_{k-1} - \mathbf{v}_k) \leq \frac{\|\mathbf{d}_k - \nabla_\mathbf{x} f(\mathbf{z}_k, \mathbf{y}_k)\|^2}{\alpha_k - L\gamma_k} \quad (22)
$$

Combining Eq.(21) and Eq.(22), we get

$$
\begin{aligned}
\mathbb{E}[f(\mathbf{x}_k, \mathbf{y}_k) - f(\tilde{\mathbf{x}}, \mathbf{y}_k)] \leq & (1-\gamma_k)\mathbb{E}[f(\mathbf{x}_{k-1}, \mathbf{y}_k) - f(\tilde{\mathbf{x}}, \mathbf{y}_k)] + \gamma_k \zeta_k \\
& + \frac{\gamma_k \alpha_k}{2}\mathbb{E}[\|\mathbf{v}_{k-1} - \tilde{\mathbf{x}}\|^2 - \|\mathbf{v}_k - \tilde{\mathbf{x}}\|^2] + \frac{\gamma_k \mathbb{E}[\|\mathbf{d}_k - \nabla_\mathbf{x} f(\mathbf{z}_k, \mathbf{y}_k)\|^2]}{2(\alpha_k - L\gamma_k)}
\end{aligned}
$$

Let $\delta_k = \mathbf{d}_k - \nabla_\mathbf{x} f(\mathbf{z}_k, \mathbf{y}_k)$, then $\mathbb{E}[\|\delta_k\|^2] \leq \sigma_k^2$. Let $\Phi(k) = k(k+1)(k+2)\mathbb{E}[f(\mathbf{x}_k, \mathbf{y}_k) - f(\tilde{\mathbf{x}}, \mathbf{y}_k)]$. According to Lemma 6, we have $\mathbb{E}[f(\mathbf{x}_k, \mathbf{y}_k) - \max_{\mathbf{y} \in \mathcal{Y}} f(\mathbf{x}_k, \mathbf{y})] \geq -\epsilon_k$. Thus, we can

obtain

$$
\begin{aligned}
\Phi(k) \leq& \Phi(k-1) + k(k-1)(k+1)\mathbb{E}[f(\mathbf{x}_{k-1},\mathbf{y}_k) - f(\mathbf{x}_{k-1},\mathbf{y}_{k-1}) - f(\tilde{\mathbf{x}},\mathbf{y}_k) + f(\tilde{\mathbf{x}},\mathbf{y}_{k-1})] \\
&+ k(k+1)(k+2)\gamma_k \left( \zeta_k + \frac{\alpha_k}{2}\mathbb{E}[\|\mathbf{v}_{k-1}-\tilde{\mathbf{x}}\|^2 - \|\mathbf{v}_k - \tilde{\mathbf{x}}\|^2] + \frac{\sigma_k^2}{2(\alpha_k - L\gamma_k)} \right) \\
\leq& \Phi(k-1) + k(k-1)(k+1)\mathbb{E}[\epsilon_{k-1} - f(\tilde{\mathbf{x}},\mathbf{y}_k) + f(\tilde{\mathbf{x}},\mathbf{y}_{k-1})] \\
&+ k(k+1)(k+2)\gamma_k \left( \zeta_k + \frac{\alpha_k}{2}\mathbb{E}[\|\mathbf{v}_{k-1}-\tilde{\mathbf{x}}\|^2 - \|\mathbf{v}_k - \tilde{\mathbf{x}}\|^2] + \frac{\sigma_k^2}{2(\alpha_k - L\gamma_k)} \right) \\
\leq& \Phi(0) + \sum_{s=1}^k s(s-1)(s+1)\epsilon_{s-1} - \sum_{s=1}^k s(s-1)(s+1)\mathbb{E}[f(\tilde{\mathbf{x}},\mathbf{y}_s) - f(\tilde{\mathbf{x}},\mathbf{y}_{s-1})] \\
&+ \sum_{s=1}^k s(s+1)(s+2)\gamma_s \left( \zeta_s + \frac{\alpha_s}{2}\mathbb{E}[\|\mathbf{v}_{s-1}-\tilde{\mathbf{x}}\|^2 - \|\mathbf{v}_s - \tilde{\mathbf{x}}\|^2] + \frac{\sigma_s^2}{2(\alpha_s - L\gamma_s)} \right) \\
=& \sum_{s=1}^k s(s-1)(s+1)\epsilon_{s-1} + 3\sum_{s=1}^{k-1} s(s+1)\mathbb{E}[f(\tilde{\mathbf{x}},\mathbf{y}_s)] - k(k-1)(k+1)\mathbb{E}[f(\tilde{\mathbf{x}},\mathbf{y}_k)] \\
&+ \sum_{s=1}^k s(s+1)(s+2)\gamma_s \left( \zeta_s + \frac{\alpha_s}{2}\mathbb{E}[\|\mathbf{v}_{s-1}-\tilde{\mathbf{x}}\|^2 - \|\mathbf{v}_s - \tilde{\mathbf{x}}\|^2] + \frac{\sigma_s^2}{2(\alpha_s - L\gamma_s)} \right)
\end{aligned}
\tag{23}
$$

According to the parameter setting, we have

$$
\sum_{s=1}^k s(s+1)(s+2)\gamma_s\zeta_s = \frac{1}{192}kLD_\mathcal{X}^2, \quad \sum_{s=1}^k s(s+1)(s+2)\frac{\gamma_s\sigma_s^2}{2(\alpha_s - L\gamma_s)} \leq kLD_\mathcal{X}^2 \tag{24}
$$

We also have

$$
\begin{aligned}
&\sum_{s=1}^k s(s+1)(s+2)\frac{\gamma_s\alpha_s}{2}(\|\mathbf{v}_{s-1}-\tilde{\mathbf{x}}\|^2 - \|\mathbf{v}_s - \tilde{\mathbf{x}}\|^2) \\
=& 9\kappa L\sum_{s=1}^k s(\|\mathbf{v}_{s-1}-\tilde{\mathbf{x}}\|^2 - \|\mathbf{v}_s - \tilde{\mathbf{x}}\|^2) \\
\leq& 9\kappa L\sum_{s=0}^{k-1} \|\mathbf{v}_s - \tilde{\mathbf{x}}\|^2 \leq 9k\kappa LD_\mathcal{X}^2.
\end{aligned}
\tag{25}
$$

Substituting Eq.(24) and Eq.(25) into Eq.(23) and using the fact that $\Phi(0) = 0$, we have

$$
\Phi(k) \leq \sum_{s=1}^k s(s-1)(s+1)\epsilon_{s-1} + 3\sum_{s=1}^{k-1} s(s+1)\mathbb{E}[f(\tilde{\mathbf{x}},\mathbf{y}_s)] - k(k-1)(k+1)\mathbb{E}[f(\tilde{\mathbf{x}},\mathbf{y}_k)] + 11k\kappa LD_\mathcal{X}^2.
$$

Then, for any $\tilde{\mathbf{y}} \in \mathcal{Y}$, we have

$$
\begin{aligned}
&\sum_{s=1}^k s(s-1)(s+1)\epsilon_{s-1} + 11k\kappa LD_\mathcal{X}^2 \\
\geq& \Phi(k) - 3\sum_{s=1}^{k-1} s(s+1)\mathbb{E}[f(\tilde{\mathbf{x}},\mathbf{y}_s)] + k(k-1)(k+1)\mathbb{E}[f(\tilde{\mathbf{x}},\mathbf{y}_k)] \\
=& k(k+1)(k+2)\mathbb{E}[f(\mathbf{x}_k,\mathbf{y}_k)] - 3\sum_{s=1}^k s(s+1)\mathbb{E}[f(\tilde{\mathbf{x}},\mathbf{y}_s)] \\
\geq& k(k+1)(k+2)\mathbb{E}[f(\mathbf{x}_k,\mathbf{y}_k) - f(\tilde{\mathbf{x}},\bar{\mathbf{y}}_k)] \\
\geq& k(k+1)(k+2)\mathbb{E}[f(\mathbf{x}_k,\tilde{\mathbf{y}}) - f(\tilde{\mathbf{x}},\bar{\mathbf{y}}_k) - \epsilon_k]
\end{aligned}
$$

where the second to last inequality is by the concavity of $f(\mathbf{x},\cdot)$ and the definition of $\bar{\mathbf{y}}_k$. Then, we can obtain the bound of the expectation of the primal-dual gap:

$$
\mathbb{E}[f(\mathbf{x}_k,\tilde{\mathbf{y}}) - f(\tilde{\mathbf{x}},\bar{\mathbf{y}}_k)] \leq \frac{1}{k(k+1)(k+2)}\left( \sum_{s=1}^k s(s+1)(s+2)\epsilon_s + 11k\kappa LD_\mathcal{X}^2 \right) \leq \frac{12\kappa LD_\mathcal{X}^2}{(k+1)(k+2)}
$$

where the last equation comes from the fact that $\epsilon_k = \frac{\kappa L D_\mathcal{X}^2}{k(k+1)(k+2)}$.

$\square$

### C.3 Proof of Corollary 3

If the objective function is of finite-sum form, the performance of iSTORC is the same as STORC. According to Corollary 2 of [11], the iSTORC algorithm requires $\mathcal{O}\left((n+\kappa^2)\log\frac{\kappa L D_\mathcal{Y}^2}{\epsilon_k}\right)$ IFO calls and $\mathcal{O}\left(\frac{\kappa L D_\mathcal{Y}^2}{\epsilon_k}\right)$ LO calls. Thus, the IFO and LO complexity of MPCGS are respectively $\mathcal{O}\left(\sum_{k=1}^{N}(P_k + \sum_{r=1}^{R}(n+\kappa^2)\log\frac{\kappa L D_\mathcal{Y}^2}{\epsilon_k})\right)$ and $\mathcal{O}\left(\sum_{k=1}^{N}\sum_{r=1}^{R}\left(\frac{\kappa L D_\mathcal{Y}^2}{\epsilon_k} + \frac{\beta_k D_\mathcal{X}^2}{\zeta_k}\right)\right)$, where $R$ is the number of iterations of Stochastic-Prox-step. Theorem 3 implies that $N$ should be the order $\Theta\left(\sqrt{\frac{\kappa L D_\mathcal{X}^2}{\epsilon}}\right)$. Plugging in all parameters obtains the complexity of MPSCGS.

### C.4 Proof of Corollary 4

Suppose the objective function has expectation form. According to Corollary 2, the the iSTORC algorithm requires $\mathcal{O}\left((\sqrt{\kappa}/(L\epsilon_k) + \kappa^2)\log\frac{\kappa L D_\mathcal{Y}^2}{\epsilon_k}\right)$ SFO calls and $\mathcal{O}\left(\frac{\kappa L D_\mathcal{Y}^2}{\epsilon_k}\right)$ LO calls. The rest analysis follows the proof of Corollary 3.