[Reviews · NeurIPS 2020]

Review 1

Summary and Contributions: This paper investigates projection-free algorithms for saddle point problems and proposes a new algorithm, by blending the conditional gradient sliding and mirror-prox for convex-strongly-concave saddle point problems. The proposed method is shown to guarantee eps-approximation after an order of 1/sqrt(eps) gradient evaluations and 1/eps^2 linear minimization oracles (excluding logarithmic factors). The paper also presents an extension of the proposed method for the stochastic setting. In this case, the (stochastic) gradient complexity also gets to O(1/eps^2). Finally, it presents numerical experiments on a robust multi-class classification to support these theoretical findings. ---- update after rebuttal ---- I read all reviews and the author feedback. It seems like all reviewers agree on the main points. (1) the conceptual novelty is limited. (2) the abstract over-claims the content, by claiming O(1/sqrt(eps)) iterations never mentions O(1/eps^2) lmo complexity and claiming to solve general convex-concave problems while the method is for convex-strongly-concave problems. (3) the paper kind of exaggerates the practical significance of the proposed approach. But I also agree that the paper provides novel rates, which can partly compensate for (1), and (2) and (3) can be fixed in the revision. I increased my score to 6.

Strengths: There are not many alternatives for the model problem studied in this paper, i.e., for saddle point problems with a convex-strongly-concave smooth function when both constraint sets X and Y are hard to project onto. The results are novel, to the best of my knowledge.

Weaknesses: The significance of the proposed method is not very clear. The proposed algorithm addresses a specific template where projection onto both sets are hard, and the objective is smooth convex-strongly-concave. I am not aware of many applications that fall into this template. Section 2.3 presents a multi-class classification problem for motivation, but this problem looks engineered specifically for the proposed algorithm. Why do we consider nuclear norm constraint for this classification problem? Is there any real application that uses this formulation? What are some other real applications that fall into the model problem template?

Correctness: I read the main text carefully. I did not verify the proofs in the appendix, but the results seem reasonable.

Clarity: The presentation is mostly clear, but some parts are lacking important details. For instance, the abstract tells that the paper introduces a new algorithm for the general convex-concave saddle point problems, but I only see results for convex-strongly-concave problems. Similarly, the abstract claims O(1/sqrt(eps)) iteration complexity, but it does not mention the fact that each iteration calls the linear minimization oracle multiple times, which increases the linear minimization oracle complexity to O(1/eps^2).

Relation to Prior Work: Related works are summarized in a subsection of half-page at the end of the introduction. This section provides a good overview of the FW-type methods for saddle point problems and their limitations. (arXiv:1804.08554, section 5.4 and 5.6) can be added.

Reproducibility: Yes

Additional Feedback: - Eqns (2) & (3): This notation might be confusing for inexperienced readers. Min and max here can be perceived as functions rather than optimization problems. To clarify, you might either put some space between minmax and f, or simply drop minmax and write f(x,y) = E[F(x,y,xi)]. You already have minmax in the model problem (1) anyway. - Line 83: It is better to mathematically specify the notion of smoothness (i.e., Lipschitz continuous gradients) since smoothness may have a different definition in some other fields of functional analysis. - Line 116: the linear optimization on Xc only needs to find the top singular vector of X, which only costs O(nnz(X)) time. This statement is inaccurate. First, we typically do not use LMO on X, but on a gradient term. Naturally, the cost depends not on the nnz(X) but nnz of its input. Second, the cost of finding the top singular vector is typically high, if you are looking for exact computation. What is used in the FW-literature is an approximate singular vector, and this needs to be clarified. And third, the cost (of approximating) is not O(nnz(X)). Roughly, O(nnz(X)) is the cost of a single iteration of the power method (or the Lanczos algorithm) that is frequently used to approximate the top singular vector. The overall cost of the power method (or Lanczos) also depends on the tolerance level, and the tolerance depends on the desired accuracy “epsilon” of the FW algorithm (or any other projection-free method). - Line 120: Typo: Nestrov -> Nesterov also at the same line … gradient decent -> gradient descent - Line 122: Typo: notices -> notice - The authors might want to reduce think about ways to reduce the preliminary sections, maybe by deferring some of the material to the appendix. In the current draft, the new material essentially starts on page 5. - Section 5 (experiments): You can (and should) also consider FW-type methods that use the LMO for X and projection for Y, since Y is a simplex and the cost of projection onto simplex is not a clear computational burden. - Line 207: Typo: our methods outperforms -> outperform


Review 2

Summary and Contributions: This paper studies convex-concave saddle-point optimization in case both variables are in convex and compact domains. it is assumed that both domain are only equipped with a linear minimization oracle (e.g., computing projection onto feasible set is computationally expensive) and so the authors develop a method that is purely projection-free and relies on the Frank-Wolfe method. The authors extend the results to the stochastic and finite-sum settings. In terms of techniques, the authors adopt the conditional-gradient-sliding method of Lan et al. and use it to simulate Nemirovski's mirror-prox method for saddles using only linear optimization steps. With this respect, there is not a highly novel *conceptual* innovation, but the overall technical effort and achievement is above the bar in my taste. *** post feedback *** I have ready the feedback and would like to keep my score.

Strengths: Overall this papers offers a solid contribution. The question of solving saddle-point problems using only projection-free methods is interesting (though I think the practical applications are more limited than the authors admit, but I still think this is a good problem to study) and the technical derivations are challenging enough and will be of interest to the community.

Weaknesses: Parts of the writing could be improved: 1. It is not clear why the assumption that the objective is strongly concave is needed. It will be beneficial if the authors comment on it in the paper. 2. It seems that the bounds are loose at several points and it was good if the authors would comment on the bound and whether they believe there is room for improvement. For example in Corr 1. the linear oracle bound scales as standard for cond. grad. but then there is also kappa^2 and instead of having something like D_x^2+D_y^2 we have D_x^2D_y^2. I strongly suspect this is suboptimal and it will be good if the authors relate to it. Also in stochastic case, since objective is strongly concave we may expect that parts of the sample complexity will only scale linearly with 1/eps but they all scale with 1/eps^2. Eq between lines 143-144: minimization over v? (not u) Line 4 of Alg 3: not clear what we get v_k here as one of the outputs of prox-step if its updated in the following line via CndG

Correctness: Yes

Clarity: Yes

Relation to Prior Work: Yes

Reproducibility: Yes

Additional Feedback:


Review 3

Summary and Contributions: This paper proposes new projection free algorithms to solve minimax problems. The idea of this work is to inexactly solve the prox step of The Mirror prox method Using the CGS method. The authors also propose a method that only requires stochastic gradients and linear oracle by using an inexact variant of STORC with increasing minibatches.

Strengths: The rates proposed are reasonable and only require LMO instead of projections. This convergence result on constrained minimax optimization only requiring LMO is novel. Theorem 2 provides an improvement over SCGS in a particular regime for \epsilon.

Weaknesses: In constrained optimization, the most expensive oracle is usually the LMO (computing gradients is usually more expensive than solving a linear program). In Theorem 1, the LMO complexity is O(1/\epsilon^2) which is quite slow for a batch method. 2) The proposed algorithms have lots of hyperparameters to tune (since the condition number is usually challenging to compute) and have an inner-outer loop structure. 3) Theorem 2 and 3 consider minibatches with increasing size 4) - L40 is a bit of an over-claim 'the first [...] algorithm for general convex-concave saddle point problem', since the theoretical results are in the convex-strongly-concave setting.

Correctness: I am a bit confused about Remark 2. since when \epsilon is small we could have \log(1/\espilon) >> \sqrt(\kappa). Moreover, isn’t the condition you would like to require (\sqrt(\kappa)/\epsilon >> \kappa^2) ? I am also concerned about the additive value of SVRG-like update in the non-finite sum setting. Usually such update scheme does not provide improvement in that case. Can you details the intuition why in that case such update would provide an improvement over SCGS ?

Clarity: Yes

Relation to Prior Work: contributions? I would like the author to address better the related work with respect to STORC [17] and SCGS [9]. Lanczos algorithm should be cited see: D. Gabay and B. Mercier. A dual algorithm for the solution of nonlinear variational problems via finite element approximation. Computers & Mathematics with Applications, 1976. J. Kuczyński and H. Woźniakowski. Estimating the largest eigenvalue by the power and Lanczos algorithms with a random start. SIAM. J. Matrix Anal. & Appl., 1992.

Reproducibility: Yes

Additional Feedback: Regarding the experiments, SVRE has no-guarantees in the convex-strongly-concave setting. However, It would If you were doing logistic regression with ell_2 regularization. In that case, the problem would be strongly-convex-strongly-concave There is probably some typos in (4): there is no labels b_i in the formulation. 1/n not necessary since y_i sums to 1. L 105: 'a solution' (LMO can have several solutions) L189 Theorem Lemma 5: it is a 4 on the denominator === UPDATE AFTER REBUTTAL === After having read the rebuttal and the other reviews I would like to emphasize that if the paper is accepted the author must fix the iteration complexity claims mentioned by R1 as well as the convex-concave claim L40. I am disappointed that the authors did not answer my question about "the additive value of SVRG-like update in the non-finite sum setting" Overall I still maintain my weak accept but do not go above.

[Author Response · NeurIPS 2020]

We greatly appreciate the reviewers' effort and helpful comments. We will fix the typos and polish the writing by incorporating the reviewers' suggestions.

**Response to Reviewer #1**

**Comment 1:** "The significance of the proposed method is not very clear..."
**Response 1:** First, the question of solving saddle-point problems using only projection-free methods is interesting (Reviewer #3 also mentions this point). It also has great theoretical significance in the optimization area.

Secondly, though our analysis is specified for the convex-strongly-concave setting, there is a simple way to adopt our algorithms to solve the general convex-concave saddle point problems. For a convex-concave function $f(\mathbf{x}, \mathbf{y})$, we can construct a convex-strongly-concave function as $f_\epsilon(\mathbf{x}, \mathbf{y}) = f(\mathbf{x}, \mathbf{y}) - \epsilon\|\mathbf{y} - \mathbf{y}_0\|^2$ and solve $f_\epsilon(\mathbf{x}, \mathbf{y})$ by our algorithms (such as MPCGS). Though the convergence rate of this method could be suboptimal, it's a practical way to deal with the general convex-concave situations.

In addition, [6] shows some examples of saddle point algorithms where projection onto the constrain sets is hard. These applications includes robust optimization, two-player games and sparse structured SVM.

**Comment 2:** "Why do we consider nuclear norm constraint for this classification problem?"
**Response 2:** Nuclear norm is a popular penalty in multi-class classification because datasets with many categories usually exhibit low-rank embedding of the classes behaviour (see [4]).

**Comment 3:** "(arXiv:1804.08554, section 5.4 and 5.6) can be added."
**Response 3:** We find that this paper does not have section 5.4 and 5.6. Also, it is irrelevant to our paper. Perhaps you give the wrong paper id.

**Comment 4:** "The presentation is mostly clear, but some parts are lacking important details."
**Response 4:** We will modify the confused sentences and clarify our results.

**Comment 5:** "Line 116: the linear optimization on Xc only needs to find the top singular vector of X, which only costs O(nnz(X)) time. This statement is inaccurate."
**Response 5:** You're right. The complexity should be $\tilde{O}(\frac{N}{\sqrt{\epsilon}})$, where $N$ is the number of non-zero entries in the gradient.

**Response to Reviewer #3**

**Comment 1:** "It is not clear why the assumption that the objective is strongly concave is needed."
**Response 1:** Notice that we adopt CGS algorithm to approximately solve a concave problem in Alg 4 (line 3). When the objective is strongly concave, the CGS method only requires to call $\sqrt{\kappa}\log(1/\epsilon)$ SFO. When the objective is not strongly concave, the CGS method requires to call $1/\sqrt{\epsilon}$ SFO. The convergence rate of CGS will significantly influence the total number of iterations of our algorithm because CGS is performed in the inner loop.

**Comment 2:** "It seems that the bounds are loose at several points."
**Response 2:** For our algorithms, we think our bounds are almost tight. We think that there exists better algorithms which only requires to call $O(1/\epsilon)$ LO as the projection-free algorithms for the convex optimization, but finding such an algorithm is a big challenge because minimax problem is much more complicate than the minimization problem.

**Comment 3:** "Line 4 of Alg 3: not clear what we get $v_k$ here as one of the outputs of prox-step if its updated in the following line via CndG"
**Response 3:** Actually, we do not compute $v_k$ via CndG. We only update $x_k, y_k$ and $v_k$ by the prox-step. According to Alg 3 (the procedure of prox-step), the results of the prox-step guarantee that $x_k, y_k$ and $v_k$ satisfies the equations and inequality in the Line 4 of Alg 3.

**Response to Reviewer #6**

**Comment 1:** "L40 is a bit of an over-claim".
**Response 1:** We will modify the over-claim sentences and clarify our setting. On the other hand, there is a simple way to adapt our methods to the convex-concave setting (see the second paragraph of the Response 1 to Reviewer #1).

**Comment 2:** "I am a bit confused about Remark 2. since when $\epsilon$ is small we could have $\log(1/\epsilon) \gg \sqrt{\kappa}$. Moreover, isn't the condition you would like to require $\sqrt{\kappa}/\epsilon \gg \kappa^2$?"
**Response 2:** The condition should be $2^{-\sqrt{\kappa}} < \epsilon < \kappa^{-1.5}$. Then we can get $(\sqrt{\kappa}/\epsilon + \kappa^2)\log(1/\epsilon) < \kappa/\epsilon$.

**Comment 3:** "SVRE has no-guarantees in the convex-strongly-concave setting."
**Response 3:** To our knowledge, there is no stochastic projection algorithm has guarantees in the convex-strongly-concave setting. On the other hand, we have already took a nuclear norm regularization. Usually it does not need additional L2 regularization.

[Meta-Review · NeurIPS 2020]

After discussing and considering the rebuttal, all reviewers agreed that this paper makes a nice contribution to NeurIPS and recommend acceptance. While the practical significance of the work is unclear, the reviewers agreed that its theoretical contribution was interesting. The reviewers made several important suggestions about the write-up that should be implemented in the camera-ready version of the paper (among others, fixing the abstract, and some of the claims). The authors should carefully implement them, as promised in their rebuttal.